# The bacterial toxin colibactin triggers prophage induction

Justin E. Silpe[1,4], Joel W. H. Wong[1,4], Siân V. Owen[2], Michael Baym[2] & Emily P. Balskus[1,3 ✉]

Colibactin is a chemically unstable small-molecule genotoxin that is produced by several different bacteria, including members of the human gut microbiome[1,2]. Although the biological activity of colibactin has been extensively investigated in mammalian systems[3], little is known about its effects on other microorganisms. Here we show that colibactin targets bacteria that contain prophages, and induces lytic development through the bacterial SOS response. DNA, added exogenously, protects bacteria from colibactin, as does expressing a colibactin resistance protein (ClbS) in non-colibactin-producing cells. The prophage-inducing effects that we observe apply broadly across different phage–bacteria systems and in complex communities. Finally, we identify bacteria that have colibactin resistance genes but lack colibactin biosynthetic genes. Many of these bacteria are infected with predicted prophages, and we show that the expression of their ClbS homologues provides immunity from colibactin-triggered induction. Our study reveals a mechanism by which colibactin production could affect microbiomes and highlights a role for microbial natural products in influencing population-level events such as phage outbreaks.

Microbial communities, including the human microbiome, are rich sources of bioactive natural products. However, the biological roles of natural products in these habitats are typically poorly understood. A bacterial natural product of particular relevance to human health is colibactin, a chemically reactive small-molecule genotoxin produced by gut bacteria that have a 54-kb hybrid nonribosomal peptide synthetase-polyketide synthase (NRPS-PKS) biosynthetic gene cluster known as the *pks* island (Fig. 1a). This gene cluster is predominantly found in human-associated strains of *Escherichia coli* that belong to phylogenetic group B2, but is also present in other human gut Enterobacteriaceae, as well as bacteria from the honey bee gut, a marine sponge and an olive tree knot[4–6]. Mechanistic studies have revealed that colibactin induces inter-strand DNA cross-links in vitro, causes cell-cycle arrest in eukaryotic cell culture and affects tumour formation in mouse models of colorectal cancer. Colibactin–DNA adducts have been detected in mammalian cells and in mice[7], and studies have identified colibactin-associated mutational signatures in cancer genomes, predominantly from colorectal cancer[8,9]. Despite its important biological activity, colibactin has eluded traditional isolation and structural elucidation. Information regarding its chemical structure has largely been derived from bioinformatic analyses and biochemical characterization[10–13]. These studies suggest that colibactin has a pseudodimeric structure, with a reactive cyclopropane warhead at each end that accounts for its characteristic DNA-alkylating ability[7,14,15] (Fig. 1a).

In contrast to its effects on eukaryotic organisms, the effect of colibactin on the surrounding microbial community remains largely unknown. Previous studies have indicated that colibactin production may cause broad shifts in the composition of the gut microbial community in mice and inhibit the growth of a subset of staphylococci[16,17].

However, exposure to colibactin did not affect the growth of the vast majority (97%) of bacterial species tested, and the mechanism that underlies these effects has remained elusive. We aimed to shed additional light on the activity of colibactin and its potential ecological roles in microbial communities by studying its effects on bacteria. To begin, we exposed a laboratory strain of non-colibactin-producing (*pks⁻*) *E. coli* (BW25113) to supernatants from overnight cultures of colibactin-producing *E. coli* (a heterologous expression strain called BAC-*pks*; hereafter *pks⁺*). Culture supernatants did not inhibit the growth of the laboratory *E. coli* strain (Extended Data Fig. 1a), in line with analogous reports in mammalian cells[1]. To test whether growth inhibition requires the presence of live colibactin-producing cells, we co-cultured *pks⁺* *E. coli* with *pks⁻* *E. coli* carrying chromosomally distinguishable markers (*lacZ*) and monitored the growth of the two populations. When started at a 1:1 ratio, the proportion of *pks⁺* *E. coli* did not change over the course of the experiment, and this outcome occurred irrespective of which strain carried the *lacZ* marker (Fig. 1b, Extended Data Fig. 1b). These results suggest that, under the conditions tested, colibactin production by one bacterium does not inhibit the growth of an isogenic, non-producing strain.

## Colibactin induces prophages

Multiple lines of evidence suggest that bacteria should be susceptible to colibactin-mediated DNA damage. For example, the final gene in the *pks* gene cluster, *clbS*, encodes a self-resistance protein that is reported to hydrolyse and destroy the reactive cyclopropane warheads of colibactin[18,19], and another gene, *clbP*, encodes a periplasmic peptidase that converts an inactive late-stage biosynthetic intermediate

[1]Department of Chemistry and Chemical Biology, Harvard University, Cambridge, MA, USA. [2]Department of Biomedical Informatics and Laboratory of Systems Pharmacology, Harvard Medical School, Boston, MA, USA. [3]Howard Hughes Medical Institute, Harvard University, Cambridge, MA, USA. [4]These authors contributed equally: Justin E. Silpe, Joel W. H. Wong. ✉e-mail: balskus@chemistry.harvard.edu

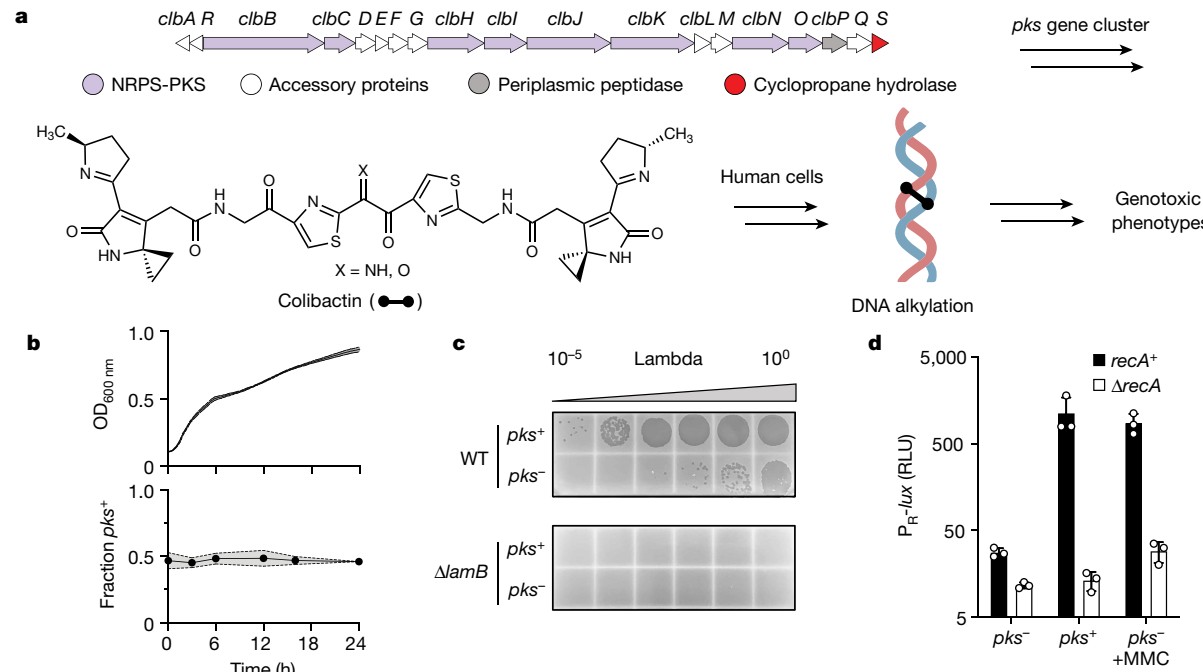

**Fig. 1 | Colibactin production specifically affects prophage-carrying bacteria. a**, Biosynthetic organization and chemical structure of the genotoxic natural product colibactin. The proposed mode of action toward human cells as a DNA-damaging agent is shown. **b**, Growth and relative abundance of *pks⁻* and *pks⁺ E. coli* in co-culture. Top, total culture density (optical density at 600 nm; $OD_{600\,nm}$) of *pks⁻lacZ⁻ E. coli* co-cultured with *pks⁺lacZ⁺ E. coli*, at a starting ratio of 1:1. Bottom, the proportion of *lacZ⁺* versus *lacZ⁻* within the same co-culture (see Extended Data Fig. 1b for swapped markers). **c**, Plaque assay obtained from 24-h co-cultures between *pks⁺* or *pks⁻ E. coli* with *E. coli*

harbouring phage lambda. Supernatants were spotted onto wild-type (WT) *E. coli* (top) and the lambda-resistant Δ*lamB* mutant (bottom). **d**, Relative light units (RLU) produced from a bioluminescent reporter encompassing the DNA-damage-inducible region of phage lambda that regulates lysis–lysogeny ($P_R$-*lux*). Reporter output measured in *recA⁺* (black) and Δ*recA* (white) *E. coli* co-cultured with *pks⁺* or *pks⁻ E. coli* in the absence or presence of MMC. RLU was calculated by dividing bioluminescence by $OD_{600\,nm}$. Data are mean ± s.d. with *n* = 3 biological replicates (**b**, **d**); or *n* = 3 biological replicates from which a single representative image is shown (**c**).

(precolibactin) to the final genotoxic metabolite in the periplasm before export[20,21]. Both bacterially encoded self-resistance mechanisms suggest that, like many toxic bacterial natural products, colibactin is potentially deleterious to non-producing bacteria. We next considered alternative consequences of colibactin-mediated DNA damage beyond inhibition of bacterial growth. One possible response of interest is phage induction. Specifically, it is known that DNA damage induced by ultraviolet irradiation or by chemical treatment (for example, mitomycin C (MMC)) activates lytic replication of prophages (a latent form of phage infection) in bacteria, killing the cell and potentially nucleating a phage epidemic within the larger microbial community[22]. We therefore wondered whether colibactin could affect bacterial populations by activating resident prophages.

To test whether colibactin production alters the behaviour of prophages in neighbouring, non-colibactin-producing lysogens, we infected wild-type *E. coli* BW25113 with phage lambda and co-cultured this lysogen with *pks⁺* or *pks⁻ E. coli*. Twenty-four hours of co-culture with the *pks⁺* strain increased phage titres by orders of magnitude above those obtained with the *pks⁻* strain (Fig. 1c). Physical separation of colibactin producers from the lysogen via a 0.4-μm filter ablated this effect (Extended Data Fig. 1c), suggesting that cell–cell contact is required[1]. Consistent with the resident prophage being the responsible agent, no plaques were observed for any condition on Δ*lamB E. coli*, which lacks the lambda phage receptor (Fig. 1c). Finally, we verified that levels of colibactin production were not markedly altered between co-culture and monoculture conditions and were unaffected by the presence of the prophage-containing strain (Extended Data Fig. 1d). These results suggest that colibactin production specifically affects prophage-carrying bacteria by inducing lytic development.

Regulation of lambda induction from the prophage state occurs via a repressor protein (cI) that is inactivated by the host-encoded SOS response, for which RecA is a master regulator[23]. To test whether prophage induction by colibactin follows a similar sequence of events, we engineered a transcriptional reporter to track the lambda lysis–lysogeny decision by fusing the lambda immunity region to the luciferase operon (*lux*) on a plasmid (hereafter called $P_R$-*lux*). Light production by luciferase therefore reports the transcriptional de-repression of phage lambda lytic replication, which is induced by known DNA-damaging-agents, such as MMC (Extended Data Fig. 1e). To examine the effect of colibactin in this system, we co-cultured *E. coli* harbouring $P_R$-*lux* with *pks⁺* or *pks⁻ E. coli*. The *pks⁺* strain induced $P_R$-*lux* in the reporter strain 40-fold compared to the *pks⁻* strain (Fig. 1d). Furthermore, the activating effect of both MMC and co-cultured *pks⁺* cells was eliminated in Δ*recA E. coli* (Fig. 1d), showing that the transcriptional de-repression requires the canonical DNA-damage-inducible SOS response. Consistent with the active genotoxin being involved, deletion of the gene encoding the late-stage biosynthetic enzyme ClbP in the producing strain abolished the activation of $P_R$-*lux* activity in co-cultured reporter cells and markedly reduced phage titres when co-cultured with the lambda lysogen (Extended Data Fig. 2a–c). Similar results were obtained using a native *pks⁺* adherent-invasive colibactin-producing *E. coli* (NC101, which is used in mouse models of colorectal cancer carcinogenesis[24]) (Extended Data Fig. 2a–c, Supplementary Discussion). Finally, addition of extracellular DNA attenuated $P_R$-*lux* activity and phage activation in a concentration-dependent, sequence-motif-specific manner (Extended Data Fig. 2d–h, Supplementary Discussion). Together, these data suggest that the ability to produce and transmit the final genotoxic product is important for the effect of colibactin on bacteria.

## pks induces human-associated prophages

Given that bacteria frequently exist in polymicrobial communities, that prophages are pervasive in these communities and that the SOS response is highly conserved, we wondered whether the genotoxic effect of colibactin could extend to prophages that reside in phylogenetically distinct, Gram-negative and Gram-positive bacteria. To investigate this, we co-cultured *pks*[+] or *pks*[−] *E. coli* with multiple isolates of prophage-carrying *Salmonella enterica* subsp. *enterica* serovar Typhimurium (*S.* Typhimurium) (one harbouring prophage P22 and another harbouring prophages BTP1 and Gifsy-1), multiple isolates of *Staphylococcus aureus* (one harbouring prophage phi11 and another harbouring phi80α), a Shiga toxin encoding isolate of *Citrobacter rodentium* (harbouring an *stx*$_{2dact}$ prophage), and a commensal isolate of *Enterococcus faecium* obtained from human faeces (harbouring a temperate phage phi1). Each phage–bacteria system resulted in a *pks*-dependent increase in prophage induction, as measured by enumerating plaque-forming units, antibody-based toxin detection and quantitative PCR (qPCR), showing that colibactin functions as a broad inducer (Fig. 2a–d). Notably, the increased Shiga toxin production from the *stx*$_{2dact}$ prophage that was observed upon co-culture of *C. rodentium* with *pks*[+] *E. coli* reveals how the effects of colibactin on susceptible bacteria and prophage have functionally relevant consequences beyond microorganisms.

Having experimentally demonstrated prophage induction in these bacteria under in vitro co-culture conditions, we next sought to test the action of colibactin in a setting that more closely resembles the complex multispecies environment of the gut. To achieve this, we anaerobically co-cultured *pks*[+] or *pks*[−] *E. coli* ex vivo with complex faecal microbiomes from C57BL/6J mice, to which we added the individual gut-associated bacteria from our above panel[25] (*S.* Typhimurium, *S. aureus*, *E. faecium* and *C. rodentium*) (Fig. 2e). In all cases except for prophage BTP1 from the *S.* Typhimurium polylysogen, we observed an increase in phage or toxin production within the *pks*[+] communities relative to the *pks*[−] communities (Fig. 2f–h). The lack of *pks*-dependent induction for BTP1—which is possibly due to the high degree of spontaneous induction in anaerobic communities—is notable given the significant induction observed under the same conditions for Gifsy-1, which is co-harboured by the same host bacterium. Together, these results demonstrate that colibactin-producing bacteria induce prophages in human- and gut-relevant strains and in complex microbial communities.

## pks[−] strains can be colibactin-resistant

Our results predict that colibactin, like MMC, is a generally effective inducer of prophages. However, unlike MMC, in which self-protection to the producing organism is thought to require the combined action of multiple resistance proteins[26,27], protection from colibactin exposure in *pks*[+] organisms involves one 170-amino-acid resistance protein, ClbS[18,19] (Fig. 1a). Studies using genetic deletions of *clbS* found that colibactin producers deficient for ClbS are viable, but that their growth depends on RecA, indicating that DNA repair mechanisms are needed for growth[18] (Extended Data Fig. 3a). Acquisition of *clbS* would therefore be a potential strategy for susceptible community members to protect themselves against the effects of colibactin. We thus wondered whether *clbS*-like genes exist in closely related, non-colibactin-producing bacteria, and whether the function of these genes may alter phage–host dynamics in response to colibactin. To gain insight into its context, we performed a bioinformatic search (tBLASTn) for genes that encode proteins with amino acid sequences identical to that of *E. coli* ClbS. In 97% of the examined hits (230 total; Supplementary Table 1), the *clbS* homologue was found in a *pks* gene cluster with the same genetic organization as that of known colibactin-producing strains. In the seven cases (3%) in which *clbS* was not associated with an intact *pks* gene cluster, the gene normally encoded upstream of *clbS*−*clbQ*−was present but truncated,

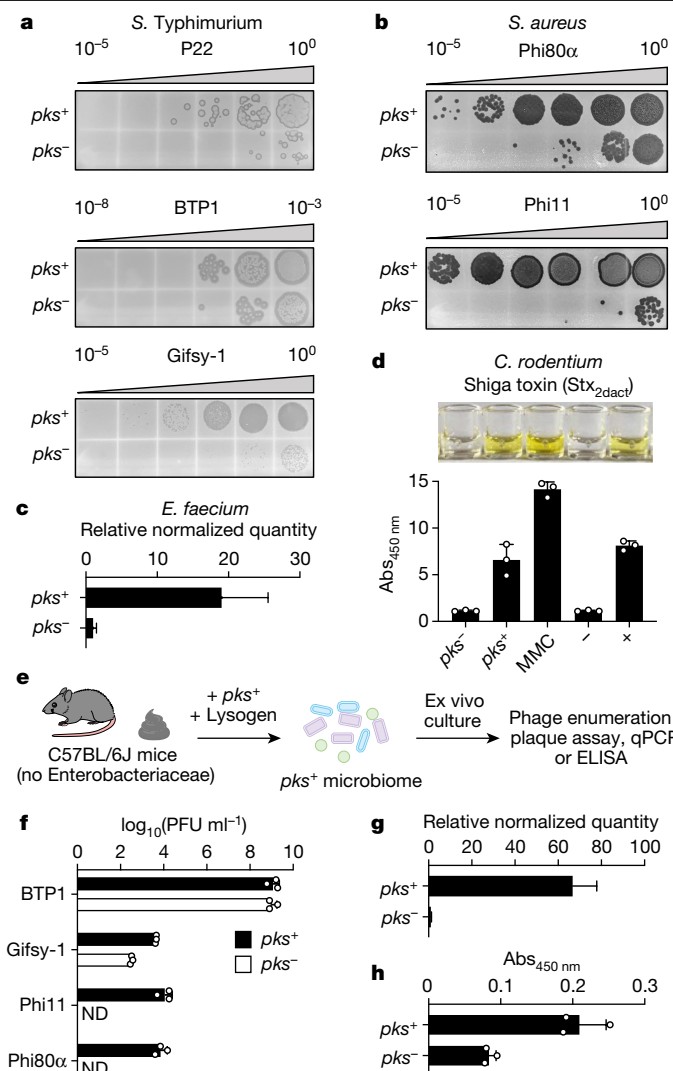

**Fig. 2 | Colibactin activates prophages in diverse bacteria and in complex communities. a**, **b**, Plaque assays obtained from co-cultures of *pks*[+] or *pks*[−] *E. coli* with a *S.* Typhimurium P22 lysogen (**a**, top), a *S.* Typhimurium BTP1 and Gifsy-1 polylysogen (**a**, bottom two panels) or two *S. aureus* lysogens (**b**). The indicator strains used for each plaque assay are specific to the different phages indicated. **c**, Relative quantities of *E. faecium*-specific host and phage DNA as measured by qPCR after co-culture with *pks*[−] or *pks*[+] *E. coli*. **d**, Enzyme-linked immunosorbent assay (ELISA) of Shiga toxin (Stx$_{2dact}$) in cultures of *C. rodentium* harbouring phage *stx*$_{2dact}$ co-cultured with *pks*[−] or *pks*[+] *E. coli* or induced with 1.5 µg ml⁻¹ MMC. Negative and positive conditions (bars designated – and +) indicate Stx-negative and Stx-positive standards included with the ELISA kit (Methods). Top, image of representative assay results from microtitre wells. Bottom, absorbance measurements from above samples (absorbance at 450 nm; Abs$_{450 nm}$). **e**, Schematic of ex vivo community experiment. Faecal communities from C57BL/6J mice were cultivated anaerobically before the addition of *pks*[−] or *pks*[+] *E. coli* along with each of the indicated phage-containing isolates indicated in **f**–**h**. Supernatants of the resulting samples were analysed for phage induction using plaque assays, qPCR and ELISA. **f**, Plaque forming units (PFU) of lysogens BTP1, Gifsy-1, phi80α and phi11 in ex vivo communities. ND, not detected. **g**, Relative quantities of *E. faecium*-specific host and phage DNA as measured by qPCR from ex vivo communities. **h**, Shiga toxin ELISA on *C. rodentium* harbouring phage *stx*$_{2dact}$ in ex vivo communities. In a, b, data are shown as a single representative image from *n* = 3 biological replicates. In **d**, **f**, **h**, data are mean ± s.d. with *n* = 3 biological replicates. In **e**, schematic created with BioRender.com. In **c**, **g**, data are mean ± s.e.m. with *n* = 3 biological replicates, each with 2 and 3 technical replicates (**c** and **g**, respectively).

and both genes were surrounded by predicted transposase-associated genes (Extended Data Fig. 3b, Supplementary Table 1), indicating that the region may be subject to horizontal transfer. This search, consistent with a recent report[28], reveals that the *clbS* gene found in *pks*+ *E. coli* also exists in isolates of the same species that lack a *pks* gene cluster.

To test whether expression of ClbS in non-colibactin-producing *E. coli* can provide protection from colibactin exposure, we introduced and expressed plasmid-encoded *clbS* (from *pks*+ *E. coli*, pTrc-*clbS*) in a *pks*− strain harbouring either the P$_R$-*lux* reporter or phage lambda (Extended Data Fig. 3c). When co-cultured with *pks*+ *E. coli*, the reporter strain harbouring the *clbS* expression vector prevented P$_R$-*lux* reporter activity, whereas the same reporter strain transformed with pTrc-Δ*clbS* did not (Extended Data Fig. 3d). ClbS did not inhibit P$_R$-*lux* reporter activity when MMC was used as the inducing agent, suggesting that protection is specific to colibactin (Extended Data Fig. 3e). Moreover, supernatants from ClbS-expressing cells did not provide protection against colibactin, indicating that ClbS-based resistance is intracellular and not shared between cells (Extended Data Fig. 3f). Consistent with the reporter activity, the lambda lysogen harbouring pTrc-*clbS* yielded 1,000-fold fewer phage particles than the same lysogen carrying pTrc-Δ*clbS* when co-cultured with *pks*+ *E. coli* (Extended Data Fig. 3g). The pTrc-*clbS* construct also repressed the induction of phage P22 when introduced into *S.* Typhimurium, indicating that this resistance mechanism is functional beyond *E. coli* (Extended Data Fig. 3g).

We next sought to examine whether *clbS*-like genes exist in more distantly related bacteria that lack the *pks* gene cluster; and, if so, whether they also have a protective function in this context. Hypothesizing that organisms that are found in close proximity with colibactin producers would benefit from colibactin resistance, we searched for more distant ClbS homologues (50% amino acid identity cut-off) in the genomes of bacteria from two specific niches in which *pks* encoders are reported to exist: the human gastrointestinal tract and the honey bee gut[5,29]. We found multiple *clbS*-like genes in *pks*− human-associated bacteria, including *Escherichia albertii* 07-3866, *Kluyvera intestini* and *Metakosakonia* sp. MRY16-398, the latter two of which were isolated from patients with gastric cancer and patients with sigmoid colon diverticulitis, respectively[30,31]. We also identified a *clbS*-like gene in *Snodgrasella alvi*, a *pks*− core member of the honey bee gut[32] (Extended Data Fig. 4a).

To assess whether these homologues could protect against colibactin-induced phage lysis, we heterologously expressed a subset of the identified ClbS-like proteins in the *E. coli* lambda lysogen or reporter strain and co-cultured these bacteria with *pks*+ *E. coli*. All four ClbS-like proteins attenuated DNA damage and prophage induction, both in terms of reporter output and plaques produced, suggesting the potential for the bacteria harbouring these genes to be protected from colibactin (Fig. 3a). Removing the niche-specific criteria and further lowering the cut-off in our search led us to uncover a wider range of ClbS-like proteins (25–80% amino acid identity relative to *E. coli* ClbS), an additional six of which we chose as a representative panel for heterologous expression and evaluation in our assays (Extended Data Fig. 4a). A summary of all ClbS-like proteins identified in our search is presented in Supplementary Table 1 (BLASTp 5,000 hits; Methods). Every ClbS-like protein tested in our panel provided protection against colibactin-induced DNA damage and prophage induction (Fig. 3a). Collectively, these results show that protection from colibactin can be gained through distantly related ClbS-like proteins that are found in bacteria that lack all other *pks* genes.

## Resistance as a phage-silencing strategy

To investigate the effect of colibactin resistance on prophage induction in non-colibactin-producing bacteria, we focused on two human-associated *clbS*+ organisms from our panel: *Metakosakonia* sp. MRY16-398 and *E. albertii* 07-3866, both of which harbour predicted DNA-damage-responsive prophages (Extended Data Fig. 4b–d,

Supplementary Discussion). In *Metakosakonia* sp. MRY16-398, one of the predicted prophages corresponds to an uncharacterized 40-kb element. We synthesized a region of approximately 1 kb from this *Metakosakonia* prophage, encompassing the putative immunity region (containing a possible cI-like repressor; Supplementary Discussion), and fused the counter-oriented promoter to *lux* on a plasmid, called P$_{Metako}$-*lux* (Fig. 3b). When introduced into *recA*+ *E. coli*, the activity of P$_{Metako}$-*lux* was activated both by MMC and by co-culture with *pks*+ *E. coli* (Fig. 3c), suggesting its DNA damage inducibility. Next, to determine whether the *clbS*-like gene encoded by *Metakosakonia* sp. MRY16-398 (*clbS*$_{Metako}$) affects induction of the *Metakosakonia* phage, we introduced plasmid-based *clbS*$_{Metako}$ into the reporter strain (resulting in P$_{Metako}$-*lux* + pTrc-*clbS*$_{Metako}$). We found that MMC continued to activate reporter expression, whereas co-culture with *pks*+ *E. coli* did not (Fig. 3c). Although we could not obtain an isolate of *Metakosakonia* MRY16-398 for these investigations, the results predict that if the ClbS protein is expressed in this *Metakosakonia* host, the organism will be resistant to the prophage-inducing effects of colibactin.

As another example, we turned to an available human-associated *clbS*+ isolate of *E. albertii* (Fig. 3d). *Escherichia albertii* 07-3866 was isolated from human faeces[33], encodes a ClbS homologue that is unique from those of *Metakosakonia* sp. MRY16-398 and *pks*+ *E. coli* (Extended Data Fig. 4a), and harbours multiple predicted prophages (Extended Data Fig. 4c). When exposed to MMC, lysates of *E. albertii* 07-3866 cultures formed distinct plaques on *E. coli*, indicating that *E. albertii* 07-3866 harbours a DNA-damage-inducible prophage (Fig. 3e). In contrast to treatment with MMC, co-culture of *E. albertii* 07-3866 with *pks*+ *E. coli* did not lead to plaque formation (Fig. 3e). We investigated whether the failure of this strain to produce phages specifically during *pks*+ co-culture could be explained by the expression of *E. albertii* ClbS. *Escherichia albertii* is related to *E. coli*, and the region that surrounds *clbS* in the *E. albertii* 07-3866 genome is highly conserved in *E. coli* (around 90% nucleotide identity in an approximately 18-kb vicinity). We transferred the *clbS* locus from *E. albertii* to the same relative location in the *pks*− *E. coli* genome. As shown in Fig. 3f, when co-cultured with *pks*+ *E. coli*, *pks*− *E. coli* harbouring the chromosomally integrated *E. albertii clbS* exhibited a reduction of approximately 50% in P$_R$-*lux* reporter activity relative to the unprotected, wild-type *clbS*− strain. The results of these experiments suggest that native *clbS* expression levels in *E. albertii* are sufficient to attenuate colibactin-specific prophage induction. More generally, our data from both *Metakosakonia* sp. and *E. albertii* lead us to propose that *clbS*+ organisms are protected from colibactin-mediated prophage induction. These results also imply that the acquisition of orphan *clbS* genes is an effective strategy for prophage-carrying bacteria to resist the production of colibactin by neighbouring community members.

## Discussion

The knowledge that colibactin induces prophages in diverse bacteria, combined with the finding that non-colibactin-producing bacteria from distinct environmental origins have functional *clbS*-like genes, leads us to speculate that colibactin production is more widespread than currently recognized, and that this genotoxin is likely to have evolved to target bacteria rather than a mammalian host. So far, studies of colibactin have primarily focused on its role in carcinogenesis, but this activity raises important questions with regard to the evolutionary role of the toxin for the producing bacterium. Colibactin genes have also been implicated in siderophore biosynthesis and microcin export, suggesting that these factors may collectively be involved in bacterial competition[34,35]. Although other functions of colibactin may exist, our discovery that it induces prophages provides one mechanism by which production of and immunity to this natural product might confer a competitive advantage over other microorganisms. For example, because cell lysis is an irreversible consequence of prophage induction[22], this mechanism could explain a previously reported observation

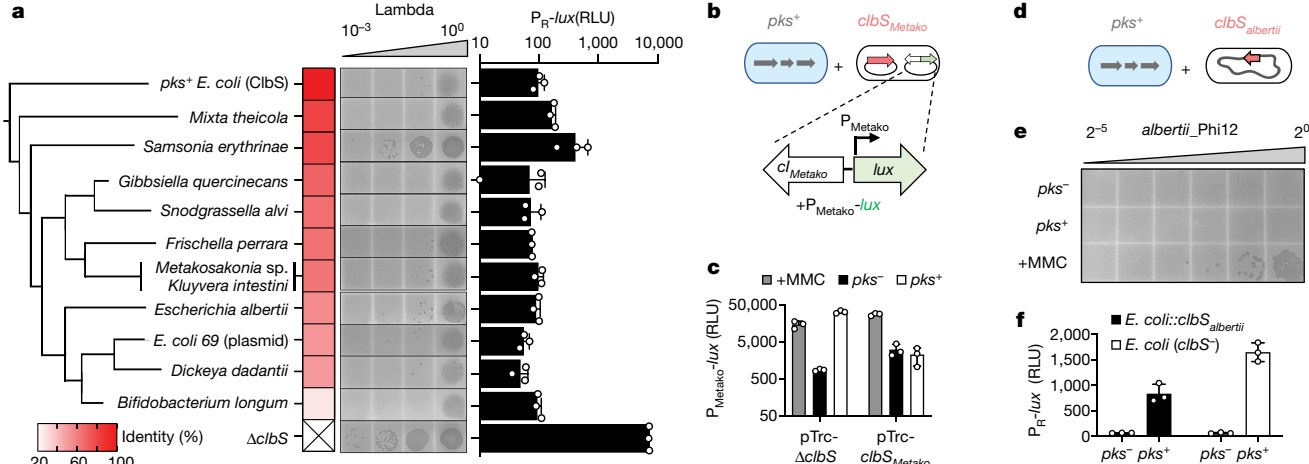

**Fig. 3 | ClbS and ClbS-like proteins from diverse bacteria provide protection against colibactin-activated prophage induction. a**, Plaque assay (images, left) and $P_R$-$lux$ reporter output (bar graph, right) obtained from $pks^+$ *E. coli* co-cultured with $pks^-$ *E. coli* harbouring a vector encoding the $clbS$-like gene of the indicated organism or the pTrc-$\Delta clbS$ vector. For plaque assays, the $pks^-$ *E. coli* harboured phage lambda; for the $P_R$-$lux$ reporter assay, the $pks^-$ *E. coli* harboured the reporter plasmid. The heat map and clustering of the ClbS-like proteins are based on amino acid identity to $pks^+$ *E. coli* ClbS. *Metakosakonia* sp. and *K. intestini* share the same ClbS sequence. **b**, Schematic of co-culture experiment between $pks^+$ *E. coli* with *E. coli* harbouring the *Metakosakonia*-derived prophage reporter ($P_{Metako}$-$lux$) and a second vector containing the $clbS$-like gene from the same organism (pTrc-$clbS_{Metako}$) or pTrc-$\Delta clbS$. **c**, $P_{Metako}$-$lux$ reporter output from co-cultures as described in **b**. Grey bars indicate the reporter response in monoculture to MMC. **d**, Schematic of co-culture experiment between $pks^+$ *E. coli* with an isolate of *E. albertii* that natively encodes a $clbS$-like gene in its genome and harbours a prophage (*albertii*_phi12). **e**, Plaque assay obtained from co-cultures as described in **d**. The supernatants were serially diluted (twofold) and spotted on *E. coli* BW25113 to measure plaque formation. **f**, $P_R$-$lux$ reporter output obtained from culturing $pks^+$ or $pks^-$ *E. coli* with $pks^-$ *E. coli* harbouring the reporter plasmid or $pks^-$ *E. coli* that was recombineered to encode $clbS_{albertii}$ from the same chromosomal locus in *E. coli* as it occurs in *E. albertii* (designated *E. coli::clbS$_{albertii}$*). In **a** (plaque assay), **e**, data are shown as a single representative image from $n = 3$ biological replicates; for **a** ($lux$ reporter), **c**, **f**, data are represented as mean ± s.d. with $n = 3$ biological replicates and RLU as in Fig. 1d.

---

of $pks$-dependent growth inhibition of a subset of *S. aureus* strains, a bacterium with an evolutionary history shaped by phage activity[17,36] (Supplementary Discussion). Moreover, the broad-spectrum activity of colibactin in inducing prophages across distinct bacteria suggests that this natural product could have effects on many members of a community, potentially accounting for colibactin-associated changes in the composition of the gut microbiome that have previously been observed in animal models[16]. Beyond bacteria, our observation that exposure to colibactin increases the production of Stx in mixed communities hints at mechanisms by which this natural product could affect host health and highlights how inducing prophages may regulate other behaviours within the microbial community.

Our study underscores major gaps in our understanding of the molecular mechanisms that underlie prophage induction in microbiomes. MMC and ultraviolet light are the most common methods of activating prophage induction in the laboratory; however, the ecologically relevant triggers for prophages found in natural environments remain largely unidentified. Previous work has shown that the human gut commensal bacterium, *Lactobacillus reuteri*, harbours a prophage that undergoes induction during gastrointestinal transit in response to dietary fructose and short-chain fatty acids[37]. In the vaginal community, metabolism of benzo[*a*]pyrene—a constituent of tobacco smoke—and subsequent secretion in the vagina induces multiple *Lactobacillus* prophages[38]. In the nasal microbial community, hydrogen-peroxide-producing *Streptococcus* has been shown to selectively eliminate prophage-carrying *S. aureus*[39]. Unlike benzo[*a*]pyrene, which humans encounter through outside exposures, and hydrogen peroxide, which has a wide range of biological targets and proposed functions, colibactin is a complex natural product produced by human gut bacteria. By uncovering the phage-inducing activity of colibactin-producing bacteria, our findings reveal a previously unrecognized mechanism by which colibactin and potentially other DNA-damaging natural products may shape microbial communities. More generally, the modulation of phage behaviours represents a distinct and underappreciated ecological role for microbial natural products. Our findings add to this growing understanding[14,40–43] and, notably, demonstrate phage induction by a natural product in co-culture. Finally, as links between the human gut virome and diseases continue to be established[44], our findings set the stage for further investigations of how gut bacterial metabolite production modulates phage behaviours and may influence human disease.

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

# Methods

## Bacterial strains, plasmids and routine cultivation

Bacterial strains and plasmids used in these studies are listed in Supplementary Table 2 and Supplementary Table 3, respectively. Unless otherwise noted, *E. coli* DH10β (NEB) was used for all strain construction and propagated aerobically in Luria-Bertani (LB-Lennox, RPI) broth at 37 °C. All experiments involving faecal communities were performed in an anaerobic chamber (70% $N_2$, 25% $CO_2$, 5% $H_2$). Oligonucleotides (Sigma) and dsDNA gene blocks (IDT) used in plasmid construction are listed in Supplementary Table 4. Plasmid construction steps and recombineering were performed using enzymes obtained from NEB (NEBuilder HiFi DNA assembly master mix, T4 DNA ligase and DpnI) and lambda red (pKD46 and pKD3), respectively. Sequencing of all inserts was performed using Sanger sequencing. Plasmid sequencing of $P_R$-*lux* revealed that the vector consists of two copies of the DNA-damage-responsive element (cI and $P_R$). Growth, reporter and lysis assays were all carried out in M9 medium supplemented with 0.4% casamino acids (M9-CAS, Quality Biological) unless otherwise specified. Antibiotics, inducers and indicators were used at the following concentrations: 100 µg ml⁻¹ ampicillin (IBI Scientific), 50 µg ml⁻¹ kanamycin (VWR), 25 µg ml⁻¹ chloramphenicol (Sigma), 100 ng ml⁻¹ MMC (Sigma), 40 µg ml⁻¹ 5-bromo-4-chloro-3-indolyl β-ᴅ-galactosidase (X-gal, Takara Bio), and 500 µM isopropyl β-ᴅ-1-thiogalactopyranoside (IPTG, Teknova), unless otherwise specified.

## Growth and competition assays

**For growth inhibition by cell-free fluids.** Overnight cultures of wild-type *E. coli* BW25113 harbouring either BAC-*pks* or the empty BAC were centrifuged (16,100*g* and 1 min) and the supernatant was passed through a 0.22-µm filter (Corning Spin-X). Growth of non-colibactin-producing *E. coli* cultures was assayed in fresh LB in the presence of varying amounts of each supernatant (5%, 10%, 20%, 50% v/v). $OD_{600}$ was measured at regular intervals using a BioTek Synergy HTX multi-mode plate-reader.

**For testing ClbS protection from cell-free fluids.** Overnight cultures of wild-type *E. coli* BW25113 harbouring either pTrc-*clbS* or pTrc-Δ*clbS* were centrifuged (16,100*g* and 1 min) before addition at 10% v/v to co-cultures containing a 1:1 ratio of *E. coli* BW25113 harbouring the $P_R$-*lux* reporter and *E. coli* BW25113 harbouring either BAC-*pks* or the empty BAC. Bioluminescence was measured after 24 h and quantified in a plate-reader as outlined below (see '*E. coli*-based reporter assay').

**For *E. coli*–*E. coli* competition assays.** Overnight cultures of *lacZ*⁺ *E. coli* MG1655 (KIlacZ, Addgene: 52696) harbouring BAC-*pks* were back-diluted 1:100 into fresh M9-CAS and mixed in a 1:1 ratio with a similarly back-diluted culture of *lacZ*⁻ *E. coli* MG1655 (delta-Z, Addgene: 52706) harbouring the empty BAC. The co-cultures were incubated at 37 °C, and, at regular intervals, an aliquot was taken for differential plating on LB supplemented with X-gal and IPTG. Both BAC combinations (*pks*⁺ versus empty) and marker combinations (*lacZ*⁺ versus *lacZ*⁻) were tested to rule out the influence of carrying the *lacZ* marker.

**For assaying RecA-dependent growth of *pks*⁺Δ*clbS E. coli*.** Overnight cultures of wild-type *E. coli* BW25113 or wild-type *E. coli* DH10β, each individually harbouring BAC-*pks* or BAC-*pks*Δ*clbS*, were back-diluted 1:100 into fresh M9-CAS. The monocultures were incubated at 37 °C and the $OD_{600\,nm}$ readings were obtained after 24 h.

**For *E. coli*–*S. aureus* competition assays.** *S. aureus* RN450 lysogenic for phi80α and *S. aureus* RN450 lysogenic for phi11 were grown overnight at 37 °C in fresh brain heart infusion (BHI) medium, and *E. coli* BW25113 harbouring BAC-*pks* or empty BAC were grown overnight at 37 °C in fresh LB broth supplemented with chloramphenicol. The overnight cultures were back-diluted 1:100 into fresh BHI medium and mixed in a 1:1 ratio and incubated at 37 °C for 24 h. The cultures were plated on LB agar supplemented with Cm for *E. coli* colony-forming units (CFUs), and mannitol salt phenol-red agar (Sigma) for *S. aureus* CFUs.

**For differential MMC susceptibility of phage-free *S. aureus* and *E. coli*.** *lacZ*⁻ *E. coli* MG1655 (delta-Z, Addgene: 52706) and *S. aureus* RN450 were grown overnight at 37 °C in fresh LB and BHI media, respectively. The overnight cultures were back-diluted 1:100 into the same respective fresh medium and a twofold dilution series of MMC was added to achieve a final concentration ranging from 78 ng ml⁻¹ to 5,000 ng ml⁻¹. Cultures were subsequently incubated overnight at 37 °C and $OD_{600\,nm}$ readings were obtained after 24 h. Normalized $OD_{600\,nm}$ was calculated as the $OD_{600\,nm}$ at a given MMC concentration relative to the $OD_{600\,nm}$ of the same strain to which no MMC was added (defined as 100%).

## Production and isolation of phage lambda by MMC induction

An overnight culture of the lambda lysogen was back-diluted 1:100 into fresh LB and incubated at 37 °C. After reaching an $OD_{600\,nm}$ of 0.4–0.5, MMC (500 ng ml⁻¹ final concentration) was added and the cultures were returned to 37 °C for an additional 3–5 h, over which time noticeable clearing occurred. After chloroform treatment and centrifugation (16,100*g* and 1 min), the clarified lysates were filter-sterilized and stored at 4 °C before use.

## Quantification of phage induction by colibactin

***E. coli*-based reporter assay.** Overnight cultures were back-diluted 1:100 into fresh M9-CAS medium with appropriate antibiotics before being dispensed (200 µl) into white-walled 96-well plates (Corning 3610). For co-culture experiments, the two cultures were mixed 1:1 immediately after back-dilution. Monoculture controls for each strain were prepared by adding 100 µl of the back-diluted cultures to an equivalent volume of M9-CAS. For DNA interference experiments, herring sperm DNA (Promega) was used. To test DNA with varying AT richness, complementary oligonucleotide pairs (JWO-1046 and JWO-1047) and (JWO-1044 and JWO-1045) were annealed in 10 mM aqueous Tris-HCl buffer, and the resulting duplexes were added to the wells at the indicated concentrations. Plates were shaken at 37 °C and the $OD_{600\,nm}$ and bioluminescence readings were obtained after 24 h. Relative light units (RLU) were calculated by dividing the bioluminescence by the $OD_{600\,nm}$.

**Phage quantification for phages of *E. coli*, *S.* Typhimurium, *S. aureus*, *E. albertii* O7-3866 and *E. faecium* E1007.** Preparing and measuring viral titres from co-cultures with phage-infected isolates was carried out according to the identical conditions used for the reporter assays with the exception that the reporter strain was substituted for the relevant lysogen. Co-cultures with phage-infected *E. coli*, *S.* Typhimurium and *E. albertii* were conducted in M9-CAS, whereas co-cultures with phage-infected *S. aureus* and *E. faecium* were conducted in BHI as the growth medium. To prepare phage lysates, cultures were transferred after 24 h co-culture to microcentrifuge tubes and centrifuged at 16,100*g* for 1 min. The supernatant was removed and passed through a 0.22-µm filter. For phage quantification by plaque assays, supernatants were diluted logarithmically from 10⁰ to 10⁻⁵, and 10 µl spotted on top agar (preparation below) containing the relevant indicator. For *E. albertii* O7-3866 phi12, 2-fold dilutions of the supernatants instead of 10-fold were used. In the case of quantifying *S.* Typhimurium phages from faecal communities, culture supernatants were concentrated approximately 40-fold from their starting volume in protein concentrators (Pierce, 100 kDa MWCO, spin columns) before use in plaque assays. For phage quantification by qPCR, supernatants were diluted 100-fold, treated with DNase (Promega) to remove residual DNA, then boiled to release encapsidated phage DNA. Host (JWO-1120 and JWO-1121) and phage

(JWO-1116 and JWO-1117) specific primer pairs were used for PCR amplification using the Luna Universal qPCR kit (NEB) in a CFX96 real-time PCR detection system (Bio-Rad). Data were processed and analysed by comparing the relative amplification within samples of phage-specific primer pairs to host-specific primer pairs (Pfaffl method) using the Gene Expression calculator in the CFX Manager software (Bio-Rad).

**Preparation of top agar.** The indicators used to assay each phage-bacteria system were as follows: for lambda-*E. coli* and phi12-*E. albertii* (wild-type *E. coli* BW25113 or the lambda-resistant *lamB::kan* mutant); for P22-*S.* Typhimurium (*S.* Typhimurium D23580ΔΦ); for BTP1-*S.* Typhimurium (*S.* Typhimurium SNW22 D23580 ΔBTP1); for Gifsy-1-*S.* Typhimurium (*S.* Typhimurium D23580 ΔΦ Δ*waaG::aph*); for phi80α and phi11 (*S. aureus* RN450). In each case, overnight cultures of the relevant indicator strains were back-diluted 1:100 into LB (for *E. coli* and *S.* Typhimurium) or BHI (for *S. aureus*) and incubated at 37 °C. At an $OD_{600 nm}$ of 0.3–0.5, *E. coli* and *S.* Typhimurium cultures were diluted 1:10 into molten LB-agar (0.6%) supplemented with 10 mM $MgSO_4$ and 0.2% maltose and poured onto a LB-agar (1.5%) plate. For *S. aureus*, cultures were back-diluted 1:10 into molten tryptic soy agar (0.6%) supplemented with 10 mM $CaCl_2$ and poured onto a denser layer (1.5%) of the same agar.

**ELISA for $Stx_{2dact}$ detection.** Detection of $Stx_{2dact}$ from both aerobic co-cultures and faecal communities was performed using the Premier EHEC test kit, which specifically detects Shiga toxins I and II (Meridian Biosciences), following the manufacturer's instructions with the following modifications: for aerobic co-cultures, overnight monocultures of *C. rodentium* harbouring $stx_{2dact}$ were back-diluted 1:100 and co-cultured in M9-CAS at a 1:1 ratio with *E. coli* BW25113 harbouring either BAC-*pks* or the empty BAC. For faecal community experiments, anaerobic monocultures of *C. rodentium* harbouring $stx_{2dact}$ were mixed with faecal communities in BHI as described in the relevant section below. To verify toxin production in response to a known DNA-damaging agent under these conditions, MMC (1.5 µg ml$^{-1}$ final concentration) was added to aerobic cultures of exponentially growing $stx_{2dact}$-harbouring *C. rodentium* in M9-CAS. In all cases, samples collected after 24-h incubations (exact volumes detailed below) were diluted in 200 µl of diluent buffer provided by the manufacturer before addition to Stx-specific antibody-coated microwells. The use of kit-provided positive and negative controls as well as all wash and substrate addition steps were carried out exactly according to the manufacturer-supplied protocol. The stop reagent was added approximately 2–5 min after adding the final substrate to each well, at which time the images used in Fig. 3c were taken. Absorbance at 450 nm ($Abs_{450 nm}$) was measured using a plate-reader. According to the manufacturer, $Abs_{450 nm}$ values ≥ 0.180 are considered a positive test result. For aerobic experiments including MMC controls, 2 µl of culture supernatants were used as the sample input. For faecal community experiments, culture supernatants were first concentrated approximately 20-fold from the initial volume in protein concentrators (Pierce, 30 kDa MWCO, spin columns). Forty microlitres of the concentrated retentate was used as the sample input.

**Faecal sample processing.** Faecal pellets from C57BL/6J mice from the Jackson Laboratory (which lack Enterobacteriaceae and do not contain colibactin-producing organisms) were suspended in pre-reduced PBS supplemented with 0.1% L-cysteine (5% w/v), then left to stand to allow insoluble particles to settle. The supernatant was carefully removed and mixed with an equal volume of 40% glycerol. Aliquots (50 µl) of this suspension were stored at −80 °C until required.

**Ex vivo culture with faecal communities.** An aliquot of the faecal suspension prepared above was thawed and inoculated into BHI (1:100) and incubated at 37 °C in an anaerobic chamber alongside the relevant

human-associated phage-containing bacteria and *E. coli* BW25113 harbouring either BAC-*pks* or the empty BAC. After 24 h incubation, the overnight cultures were back-diluted (1:1,000) and mixed in equal proportions in fresh BHI, then incubated for a further 24 h. Phages and toxin produced from faecal communities were measured in the same assays used in two-way cultures, involving plaque assays for *S.* Typhimurium (BTP1 and Gifsy-1) and *S. aureus* (phi80α and phi11), qPCR for *E. faecium* (phi1), and Stx ELISA for *C. rodentium* ($stx_{2dact}$).

**Assaying protection by *clbS*-like open reading frames.** For reporter assays, each of the *clbS*-like open reading frames (ORFs) (or the pTrc-Δ*clbS* construct) were transformed into *E. coli* BW25113 harbouring the $P_R$-*lux* reporter. After overnight growth of each strain in monoculture, strains were back-diluted 1:100 and co-cultured in M9-CAS at a 1:1 ratio with *E. coli* BW25113 harbouring BAC-*pks*. Bioluminescence was measured after 24 h incubation at 37 °C in a plate-reader as detailed for all other *E. coli*-based reporter assays above. For measuring protection by the *clbS*-like ORFs from phage induction, the same *clbS*-like ORF-encoding constructs from the reporter assay were individually transformed into a *E. coli* BW25113 lambda lysogen. An identical dilution and co-culture procedure to that of *E. coli* BW25113 harbouring BAC-*pks* was used, after which phage production was measured by plaque assay as described in the relevant section above. To measure protection provided by a chromosomal copy of *E. albertii*-encoded *clbS* (*clbS*$_{albertii}$), the locus surrounding *clbS*$_{albertii}$ from the *E. albertii* 07-3866 genome was PCR-amplified and transferred using lambda-red recombineering into wild-type *E. coli* BW25113 (JSO-1966–1973; Supplementary Table 4). The $P_R$-*lux* reporter plasmid was introduced into the resulting strain, *E. coli::clbS*$_{albertii,}$ and measured for its ability to be induced by colibactin using the identical co-culture procedure as that used for *E. coli*-based reporter assays, as noted in the relevant section above.

## Quantification of *N*-myristoyl-D-asparagine prodrug production by *pks*$^+$ *E. coli*

**For culture conditions and sample preparation.** Overnight cultures of *E. coli* BW25113 harbouring either BAC-*pks* or the empty BAC were back-diluted 1:100 and co-cultured in M9-CAS at a 1:1 ratio with phage-free *E. coli* BW25113 or lambda-infected BW25113. Cultures (1 ml) were dispensed into deep-well plates (VWR) and incubated with shaking at 37 °C. After 24 h, 10 µl deuterated (d27) *N*-myristoyl-D-asparagine (10 µM in DMSO stock solution) was added to each sample. Samples were flash-frozen in liquid nitrogen, lyophilized for 48 h, then reconstituted in methanol (1 ml) and vortexed for 1 min. Three hundred microlitres of the mixture was filtered through a 0.22 µm filter (Pall) before mass spectrometry analysis.

**For prodrug quantification.** Analysis of the *N*-myristoyl-D-asparagine prodrug in samples was performed using an ultra-high performance liquid chromatography tandem mass spectrometry (UHPLC–MS/MS) system model Xevo TQ-S (Waters). The mass spectrometer system consists of a triple quadrupole equipped with a dual-spray electrospray ionization (ESI) source. Samples were analysed using an Agilent Poroshell 120 EC-C18 column (2.7 mm, 4.6 mm × 50 mm) with the following elution conditions: isocratic hold at 90% solvent A in solvent B for 0.5 min: linear gradient from 90% to 5% solvent A in solvent B from 0.5–2 min; isocratic hold at 5% solvent A from 2–3 min, gradient from 5% to 98% solvent A in solvent B from 3–3.5 min; isocratic hold at 98% solvent A in solvent B from 3.5–4 min (solvent A: 95% water + 5% methanol + 0.03% ammonium hydroxide; solvent B: 80% isopropanol + 15% methanol + 5% water; flow rate = 0.75 ml min$^{-1}$; injection volume = 5 µl). The mass spectrometer was run in negative-mode MRM with a Cone voltage of 50 V, monitoring transitions of $m/z$ 341 -> $m/z$ 114 (retention time (rt) = 2.2 min, collision energy (CE) = 24 V) for the prodrug scaffold and $m/z$ 368 -> m/z 114 (rt = 2.2 min, CE = 28 V) for the deuterated

internal standard (d27-*N*-myristoyl-D-asparagine). For all samples, peak areas for the $m/z$ 341 -> $m/z$ 114 were normalized to the $m/z$ 368 -> $m/z$ 114 transition for the same sample, and then normalized values compared to a standard curve of unlabelled *N*-myristoyl-D-asparagine containing 100 nM d27- *N*-myristoyl-D-asparagine, which was run in triplicate.

## Bioinformatic analyses

NCBI tBLASTn (nr/nt database, expect threshold = 0.05, word size = 6, BLOSUM62 matrix) was used to identify *clbS* genes that match *E. coli* ClbS (WP_000290498) but that are found outside of *pks* clusters. The more distantly related ClbS-like proteins examined in this study (Fig. 3d) were compiled from BLASTp results using *E. coli* ClbS as the query (nr protein sequences database, expect threshold = 0.05, word size = 6, BLOSUM62 matrix, 5,000 entries). After excluding entries that occur in genomes with *pks* clusters, the isolation source of the remaining hits was considered in identifying bee gut and human-associated isolates. Other members in the representative panel selected for cloning and heterologous expression were chosen heuristically and to cover the range in per cent identities returned by the BLAST search (spanning *Mixta theicola* having 80% pairwise identity and *Bifidobacterium longum* with 26.8% pairwise identity to *E. coli* ClbS). The genomes encoding *clbS*-like genes in the representative panel were submitted to PHASTER for identification of prophage regions. Genes encoded by predicted intact prophages (score higher than 90) were further analysed by domain analysis (InterPro) for features matching the lambda repressor (DNA-binding and peptidase domains), as mentioned in the main text and Supplementary Discussion, and shown in Extended Data Fig. 4d.

## Quantification and statistical analysis

Software used to collect and analyse data generated in this study consisted of: GraphPad Prism 9 for analysis of growth- and reporter-based experiments; Gen5 v.3 for collection of growth- and reporter-based experiments; Bio-Rad CFX Manager 3.0 for quantification and analysis of qPCR data; ImageJ 1.53c for colony counting in competition experiments; and Geneious Prime 2020 for analysis of publicly available data and primer design. Data are presented as mean ± s.d. unless otherwise indicated in the figure legends. The number of independent biological replicates for each experiment is indicated for each experiment and included in the legend.

## Reporting summary

Further information on research design is available in the Nature Research Reporting Summary linked to this paper.

## Data availability

All unprocessed plaque assay images (Figs. 1c, 2a, b, 3a, Extended Data Figs. 1c, 2c, f, h, 3g) and source data (Figs. 1b, d, 2c, d, f–h, 3a, c, Extended Data Figs. 1a, b, d, e, 2b, e, g, 3a, d–f, 4b, c) generated in the course of this study are available without restriction and deposited on Zenodo (https://doi.org/10.5281/zenodo.4683077). Total PFU data are available in Supplementary Table 5, and total CFU and growth data are available in Supplementary Table 6 (Supplementary Fig. 1 in the Supplementary Discussion). Identifiers for all entries in NCBI BLAST results are listed in Supplementary Table 1. Protein accession numbers for the relevant ClbS sequences tested in this study are as follows: *E. coli* CFT073 (WP_000290498), *M. theicola* (PNS10644), *S. erythrinae* (WP_132453050), *Gibbsiella quercinecans* (WP_095844971), *S. alvi* (WP_025331471), *F. perrara* (WP_039103908), *Metakosakonia* sp. (BBE76153), *K. intestini* (PWF54517), *E. albertii* (WP_000115842), *E. coli* 69 (QDM73539), *Dickeya dadantii* (WP_038909824) and *B. longum* (WP_193641739). Accession numbers used for the design of qPCR primers and reporter construction are: *E. faecium* E1007 (AHWP00000000), *E. coli* BW25113 and lambda (NZ_CP009273 and NC_001416.1), *E. albertii* 07-3866 (NZ_CP030781) and *Metakosakonia* sp. (AP018756). Accession and identifier information can be found at NCBI. Source data are provided with this paper.

**Acknowledgements** We thank all members of the E.P.B. laboratory for insightful discussions; K. Papenfort for feedback; the laboratory of J. R. Penadés (Imperial College London) for sharing *S. aureus* strains; the laboratory of J. M. Leong (Tufts University) for sharing *C. rodentium*; and the laboratory of W. Garrett (Harvard School of Public Health) for sharing of faecal pellets. This work was supported by the National Institutes of Health (NIH) grant R01 CA208834. J.W.H.W. was supported by the A*STAR NSS (PhD) predoctoral fellowship. S.V.O. and M.B. were partially supported by the NIH NIGMS award R35GM133700, the David and Lucile Packard Foundation, and the Pew Charitable Trusts. The content is solely the responsibility of the authors and does not necessarily represent the official views of the funders.

**Author contributions** J.E.S., J.W.H.W. and E.P.B. conceived the project. J.E.S. and J.W.H.W. contributed equally to this work and ordering of authorship was determined in no particular order. J.E.S. and J.W.H.W. constructed strains, conducted all bioinformatic analyses and performed all growth-based-, reporter-based- and plaque assays. S.V.O. assisted in constructing and acquiring *S*. Typhimurium and *S. aureus* strains. All authors interpreted data, provided critical feedback and wrote the paper.

**Competing interests** The authors declare no competing interests.

**Additional information**
**Correspondence and requests for materials** should be addressed to Emily P. Balskus.

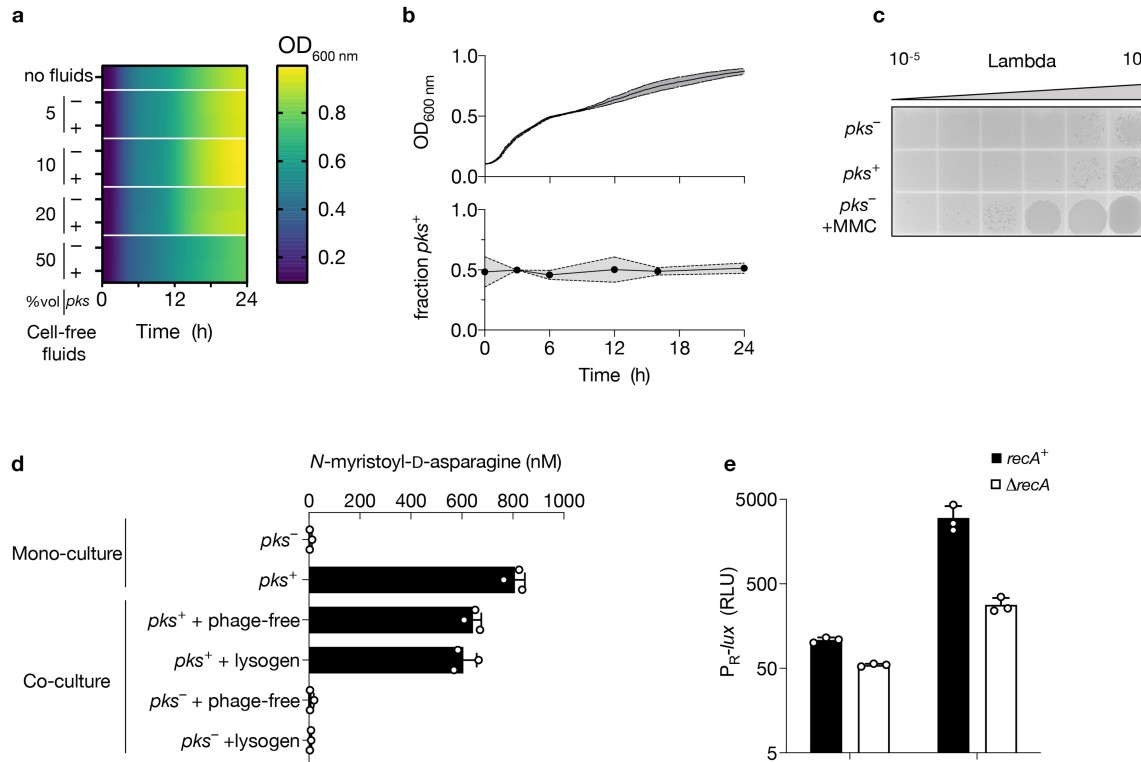

**Extended Data Fig. 1 | Colibactin production does not generally inhibit bacterial growth but induces DNA damage. a**, Growth of *pks⁻ E. coli* grown in the presence of the indicated relative volume of cell-free fluids from overnight cultures of *pks⁺ E. coli*, *pks⁻ E. coli* or without cell-free fluids added (top row). **b**, Growth and frequency of *pks⁻* and *pks⁺ E. coli* in co-culture as in Fig. 1b but with the *pks* and *lacZ* combination swapped. Upper, total culture density of *pks⁻ lacZ⁺ E. coli* co-cultured with *pks⁺ lacZ⁻ E. coli* at a starting ratio of 1:1; lower, the proportion of *lacZ⁺* versus *lacZ⁻* within the same co-culture based on differential blue-white plating over time. **c**, Plaque assay obtained after co-culturing *pks⁺* or *pks⁻ E. coli* with a lambda lysogen separated by a 0.4 µm

membrane. Where indicated, MMC was added to the opposing side of the membrane from the lambda lysogen. **d**, Concentration of the colibactin prodrug motif *N*-myristoyl-D-asparagine obtained from *pks⁻* or *pks⁺ E. coli* in monoculture and in co-culture with lysogenic and non-lysogenic (phage-free) *E. coli*. **e**, P$_R$-*lux* output in *recA⁺* (black) and Δ*recA* (white) *E. coli* harbouring the reporter plasmid in the absence and presence of MMC. For **e**, RLU as in Fig. 1d. Data represented as mean of n = 3 biological replicates (**a**), as mean ± s.d. with n = 3 biological replicates (**b**, **d**, **e**), or n = 3 biological replicates from which a single representative image is shown (**c**).

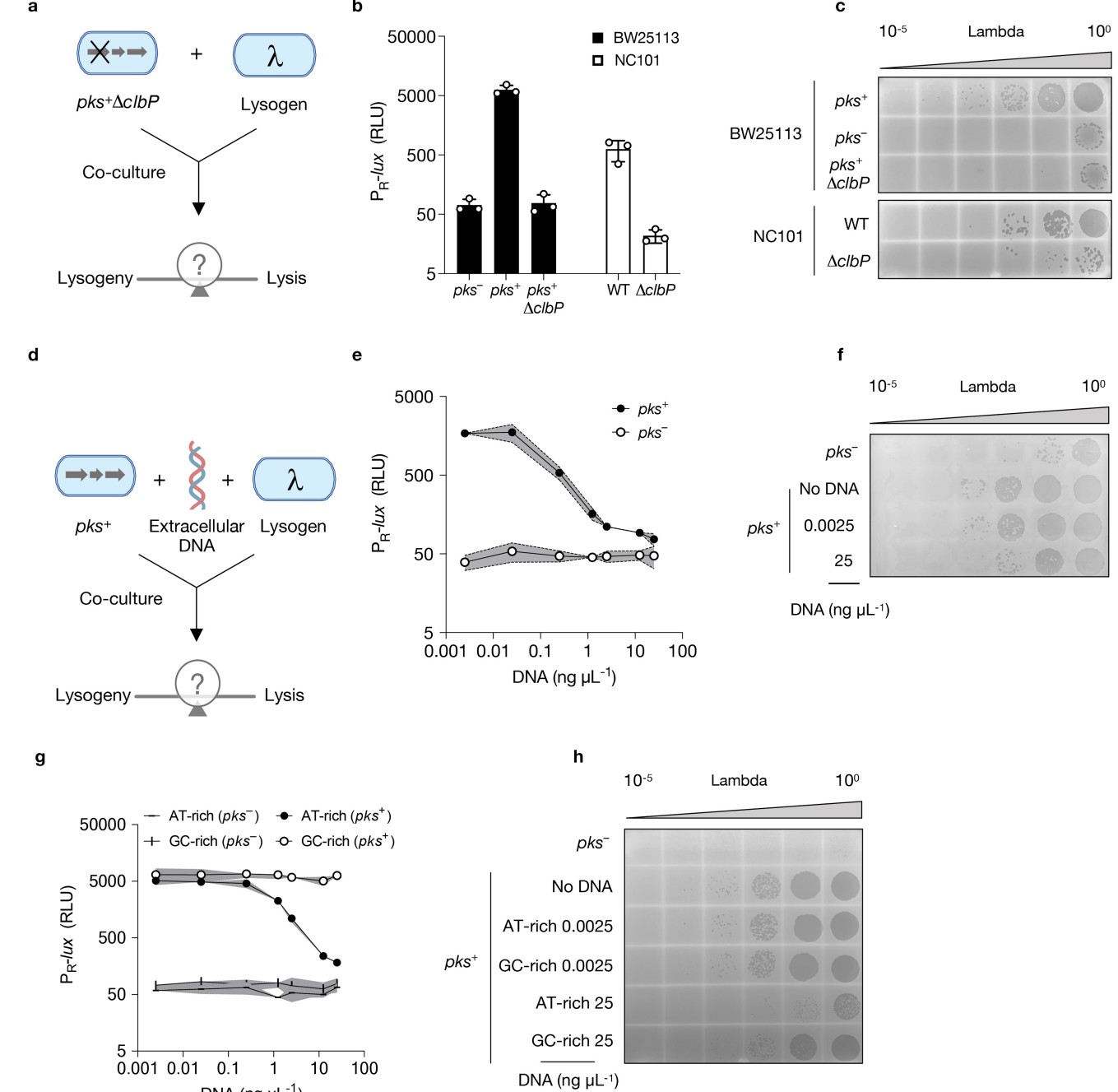

**Extended Data Fig. 2 | Prophage induction is dependent on colibactin-mediated DNA alkylation and addition of extracellular DNA ameliorates this effect. a**, Schematic of co-culture experiment with a colibactin biosynthesis-defective Δ(*clbP*) *pks* strain. **b**, P$_R$-*lux* output from reporter cells co-cultured with *E. coli* BW25113 (*pks*⁺, *pks*⁻, and *pks*⁺Δ*clbP*; black bars) or native-colibactin producing *E. coli*, NC101 (WT and Δ*clbP*; white bars). **c**, Plaque assays obtained from analogous incubations as in **b** but with a lambda lysogen used in place of the reporter strain. **d**, Schematic of co-culture experiment in which extracellular DNA is added to the medium. **e**, P$_R$-*lux* output from reporter cells co-cultured with either *pks*⁺ or *pks*⁻ *E. coli* and the indicated concentration of herring sperm DNA. **f**, Plaque assays of the analogous incubations as in **e** but with a lambda lysogen used in place of the reporter strain. **g**, P$_R$-*lux* output from reporter cells co-cultured with *pks*⁺ *E. coli* in the presence of varying amounts of extracellular DNA (AT-rich and GC-rich DNA, black and white symbols, respectively). **h**, Plaque assays of the analogous incubations as in **g** but with a lambda lysogen used in place of the reporter strain. In **b**, **e**, **g**, RLU as in Fig. 1d. Data represented as mean ± s.d. with n = 3 biological replicates (**b**, **e**, **g**); or n = 3 biological replicates from which a single representative image is shown (**c**, **f**, **h**).

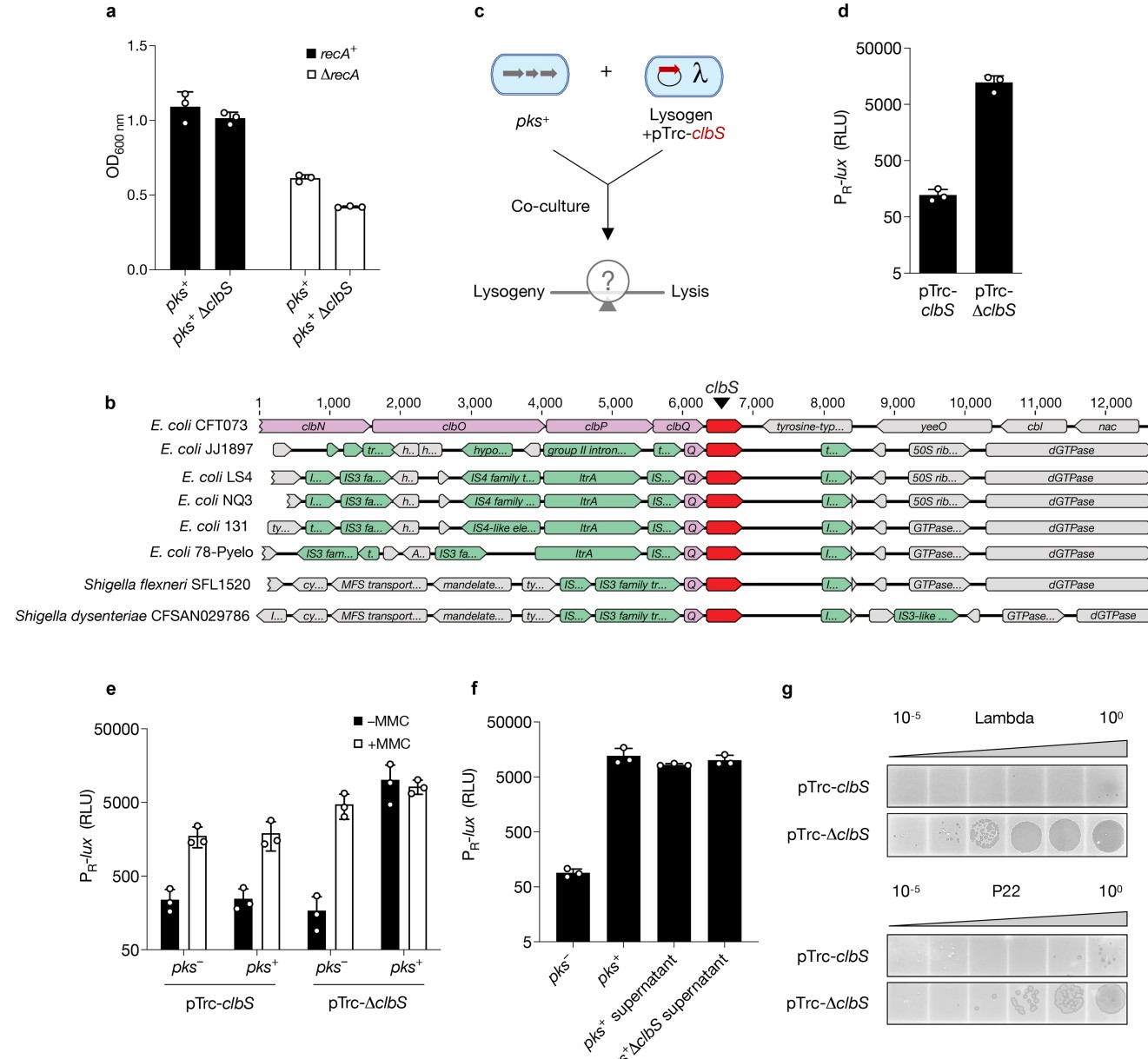

**Extended Data Fig. 3 | ClbS provides intracellular protection from colibactin, and *clbS*-like genes are present in the genomes of diverse bacteria, including those that lack *pks*-biosynthetic genes. a**, 24 h growth of *recA⁺ E. coli* (BW25113, black bars) or Δ*recA E. coli* (DH10β, white bars), each harbouring either the full *pks* cluster (*pks⁺*) or the cluster with *clbS* removed (*pks⁺*Δ*clbS*), as indicated. **b**, Genomic context of *clbS* found within the *E. coli pks* cluster encoded by a known colibactin-producing isolate (CFT073) as compared to *pks⁻* isolates of *E. coli* that lack the colibactin biosynthetic genes but contain an identical *clbS* coding sequence (red) and truncated *clbQ* (purple) in regions flanked with predicted transposase-associated genes (green-coloured genes). Numbering above genomes denotes prophage genome size in base pairs. **c**, Schematic of co-culture experiment with the gene encoding colibactin resistance, *clbS*, expressed *in trans*. **d**, $P_R$-*lux* reporter output obtained from *pks⁺ E. coli* co-cultured with *pks⁻ E. coli* harbouring the

reporter plasmid and either pTrc-*clbS* or the same vector with *clbS* removed (pTrc-Δ*clbS*) expressed *in trans*. **e**, $P_R$-*lux* reporter output in the absence and presence of MMC in *E. coli* harbouring the $P_R$-*lux* reporter plasmid, the indicated second plasmid (pTrc-*clbS* or pTrc-Δ*clbS*), and co-cultured with *pks⁻* or *pks⁺ E. coli*. **f**, $P_R$-*lux* reporter output obtained from culturing *pks⁺* or *pks⁻ E. coli* with *pks⁻ E. coli* harbouring the reporter plasmid to which cell-free supernatants of cells expressing *clbS* or a vector control (Δ*clbS*) were added (right two bars). **g**, Upper: Plaque assays obtained from analogous incubations as in **d** but with a lambda lysogen used in place of the reporter strain. Lower: Plaque assays obtained from co-culturing *pks⁺ E. coli* with *S*. Typhimurium harbouring P22 and either pTrc-*clbS* or pTrc-Δ*clbS* expressed *in trans*. In **d**, **e** and **f**, RLU as in Fig. 1d. Data represented as mean ± s.d. with n = 3 biological replicates (**a**, **d**, **e**, **f**); or n = 3 biological replicates from which a single representative image is shown (**g**).

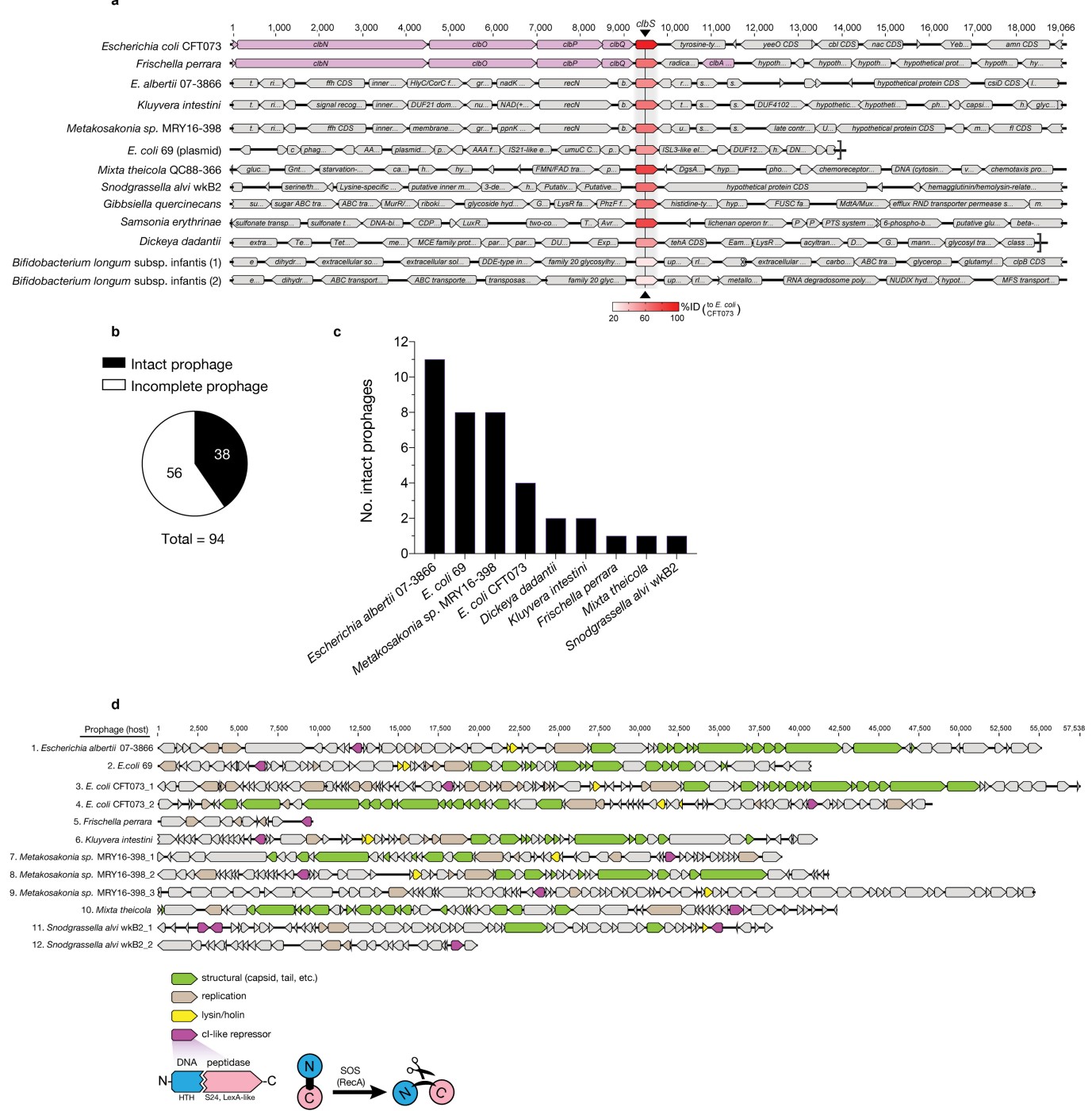

**Extended Data Fig. 4 | Prophages with predicted DNA-damage-responsive repressors co-occur in *clbS*-encoding bacteria. a**, Genomic organization surrounding *clbS*-like genes encoded by diverse bacteria identified in this study. Purple-coloured genes denote the known *pks* biosynthetic genes. *E. coli* CFT073 and *F. perrara* were previously known to carry *pks*-associated *clbS*. Red-coloured genes denote *clbS*. The saturation of red for each *clbS* is proportional to the percent identity in amino acid sequence of that gene relative to *pks*+ *E. coli* (CFT073), as indicated in the key. **b**, Distribution of PHASTER-predicted prophage regions present in the 12 bacterial genomes that encode the *clbS*-like genes tested in Fig. 3a (genomic context for each shown in a). A total of 94 prophage regions were predicted, 38 of which are considered to be intact prophages. **c**, Number and distribution of intact prophages within each

bacterial species from **b**. **d**, Organization of predicted intact prophages that encode prototypical DNA-damage-responsive repressors (12 from the 38 intact phages identified in **a** and **b**). Genes coloured according to predicted function, designated in the key. In a and d, numbering above genomes denotes size in base pairs. In d, domain analysis was used to predict the *cI*-like repressor (maroon genes) on the basis that it harbours a helix-turn-helix DNA-binding domain (blue, N-terminal domain) and a LexA-like, S24 peptidase domain (pink, C-terminal domain). The same two-domain architecture is found in the lambda cI repressor protein and confers an autoproteolytic mechanism in which the repressor is cleaved in the presence of a DNA-damage-induced, RecA-active protein complex, leading to phage lysis.

# Reporting Summary

Nature Research wishes to improve the reproducibility of the work that we publish. This form provides structure for consistency and transparency in reporting. For further information on Nature Research policies, see our Editorial Policies and the Editorial Policy Checklist.

## Statistics

For all statistical analyses, confirm that the following items are present in the figure legend, table legend, main text, or Methods section.

| n/a | Confirmed | |
|---|---|---|
| ☐ | ☒ | The exact sample size (*n*) for each experimental group/condition, given as a discrete number and unit of measurement |
| ☐ | ☒ | A statement on whether measurements were taken from distinct samples or whether the same sample was measured repeatedly |
| ☒ | ☐ | The statistical test(s) used AND whether they are one- or two-sided<br>*Only common tests should be described solely by name; describe more complex techniques in the Methods section.* |
| ☒ | ☐ | A description of all covariates tested |
| ☐ | ☒ | A description of any assumptions or corrections, such as tests of normality and adjustment for multiple comparisons |
| ☐ | ☒ | A full description of the statistical parameters including central tendency (e.g. means) or other basic estimates (e.g. regression coefficient) AND variation (e.g. standard deviation) or associated estimates of uncertainty (e.g. confidence intervals) |
| ☒ | ☐ | For null hypothesis testing, the test statistic (e.g. *F*, *t*, *r*) with confidence intervals, effect sizes, degrees of freedom and *P* value noted<br>*Give P values as exact values whenever suitable.* |
| ☒ | ☐ | For Bayesian analysis, information on the choice of priors and Markov chain Monte Carlo settings |
| ☒ | ☐ | For hierarchical and complex designs, identification of the appropriate level for tests and full reporting of outcomes |
| ☒ | ☐ | Estimates of effect sizes (e.g. Cohen's *d*, Pearson's *r*), indicating how they were calculated |

*Our web collection on statistics for biologists contains articles on many of the points above.*

## Software and code

Policy information about availability of computer code

| Data collection | Gen5 v3 for collection of of growth- and reporter-based experiments and Bio-Rad CFX Manager 3.0 for collection of qPCR based experiments. |
|---|---|
| Data analysis | Software used to analyze the data generated in this study consisted of: GraphPad Prism 9 for analysis of growth- and reporter-based experiments; Bio-Rad CFX Manager 3.0 for quantification and analysis of qPCR data; ImageJ 1.53c for colony counting in competition experiments; and Geneious Prime 2020 for analysis of publicly available data and primer design. Data are presented as the mean ± std unless otherwise indicated in the figure legends. The number of independent biological replicates for each experiment is indicated for each experiment and included in the legend. |

For manuscripts utilizing custom algorithms or software that are central to the research but not yet described in published literature, software must be made available to editors and reviewers. We strongly encourage code deposition in a community repository (e.g. GitHub). See the Nature Research guidelines for submitting code & software for further information.

## Data

Policy information about availability of data

All manuscripts must include a data availability statement. This statement should provide the following information, where applicable:
- Accession codes, unique identifiers, or web links for publicly available datasets
- A list of figures that have associated raw data
- A description of any restrictions on data availability

All unprocessed plaque assay images (Figures 1c, 2a, 2b, 3a and Extended Data Figures 1c, 2c, 2f, 2h, 3g) and source data (Figures 1b, 1d, 2c, 2d, 2f, 2g, 2h, 3a, 3c, and Extended Data Figures 1a, 1b, 1d, 1e, 2b, 2e, 2g, 3a, 3d, 3e, 3f, 4b, 4c) generated in the course of this study are available without restriction and deposited on Zenodo (doi: 10.5281/zenodo.4683078). Total PFU, and CFU and growth data (Supplemental Discussion Figure 1) are available in Supplementary Tables 5 and 6, respectively. Identifiers for all entries in NCBI BLAST results are listed in Supplementary Table 1. Protein accession numbers for the relevant ClbS sequences tested in

# Field-specific reporting

Please select the one below that is the best fit for your research. If you are not sure, read the appropriate sections before making your selection.

☒ Life sciences ☐ Behavioural & social sciences ☐ Ecological, evolutionary & environmental sciences

For a reference copy of the document with all sections, see nature.com/documents/nr-reporting-summary-flat.pdf

# Life sciences study design

All studies must disclose on these points even when the disclosure is negative.

| | |
|---|---|
| Sample size | Sample size was chosen as three biological replicates, matching the standard in the microbiology field [e.g., Erez and Steinberger-Levy et al. Nature 541, 488–493 (2017)]. All datapoints displayed in this study are available in the source data for others to access and analyze. Means and standard deviations are plotted; no additional statistical analyses were performed [see D.L. Vaux "Know when your numbers are significant. Nature 492, 180–181 (2012)]. |
| Data exclusions | No data were excluded. |
| Replication | At least three replicates were used for each experiment. All data points were plotted and are available in the source data file. No data were excluded, and all replicates were therefore considered "successful" measurements. |
| Randomization | Randomization was not formally implemented in this study, however, the choice of wells and positioning of culture tubes used in any given experiment was not pre-assigned and was therefore chosen randomly at the time of setup. |
| Blinding | Blinding was not formally applied in this study. The investigators setting up the assays also analyzed the data. The strains used in each experiment were assigned numbers that were cross-checked with the corresponding sample names/treatments only at the time the data were plotted. |

# Reporting for specific materials, systems and methods

We require information from authors about some types of materials, experimental systems and methods used in many studies. Here, indicate whether each material, system or method listed is relevant to your study. If you are not sure if a list item applies to your research, read the appropriate section before selecting a response.

## Materials & experimental systems

| n/a | Involved in the study |
|---|---|
| ☒ | ☐ Antibodies |
| ☒ | ☐ Eukaryotic cell lines |
| ☒ | ☐ Palaeontology and archaeology |
| ☐ | ☒ Animals and other organisms |
| ☒ | ☐ Human research participants |
| ☒ | ☐ Clinical data |
| ☒ | ☐ Dual use research of concern |

## Methods

| n/a | Involved in the study |
|---|---|
| ☒ | ☐ ChIP-seq |
| ☒ | ☐ Flow cytometry |
| ☒ | ☐ MRI-based neuroimaging |

## Animals and other organisms

Policy information about studies involving animals; ARRIVE guidelines recommended for reporting animal research

| | |
|---|---|
| Laboratory animals | No laboratory animals were directly used in this study. |
| Wild animals | No wild animals were used in this study. |
| Field-collected samples | No new field-collected samples were collected in this study. Bacteriophages and bacteria were used from previously published sources, listed in the strain table. |
| Ethics oversight | No ethical approval or guidance was required for this study. |

Note that full information on the approval of the study protocol must also be provided in the manuscript.

