## [Peer Review File · Nature]

Manuscript Title: The bacterial toxin colibactin triggers prophage induction

Reviewer Comments & Author Rebuttals

Reviewer Reports on the Initial Version:

Referee #1 (Remarks to the Author):

Here, Silpe et al. implicate colibactin, an NRPS/PKS molecule produced by certain *E. coli*, in prophage activation within bacteria. This work potentially provides a mechanistic basis for the previously reported capacity of colibactin production to impact the gut microbiome bacterial community and act in an antibacterial manner. Such a role for colibactin is intriguing, in that it may provide an explanation for the production of a molecule previously found to act as a mutagenic agent of colonized humans and mice. The study is straightforward, the manuscript is well written, and the experiments are easily interpreted. My major concerns relate to the narrow scope of the work.

Major concerns

(1) Silpe et al. rely entirely on *in vitro* co-culture experiments in order to investigate prophage induction and antibacterial activity of colibactin on other bacteria. I find their data convincing that under such conditions, colibactin leads to prophage induction. This is not surprising given its known non-specific DNA crosslinking activity and the well characterized induction of prophage by such insults. However, there are many reasons why this activity might not be observed (or relevant) in a more natural habitat, including the gut microbiome. For instance, colibactin activity is dependent upon highly electrophilic warhead structures that are readily quenched by extracellular molecules. Demonstrating prophage induction while measuring other potential concomitant consequences of colibactin on co-inhabiting bacteria in the context of an experiment involving a natural or model multispecies gut community is necessary in order to implicate the molecule in a meaningful way in bacterial interactions.

(2) One interesting aspect of this study is the identification of *clbS* genes in *pks*- bacteria. Silpe et al. show that these genes, when over-expressed in *E. coli*, inhibit prophage induction. However, it remains unresolved whether these genes are able to perform this function when produced natively by the bacteria that harbor them and under conditions like those I mention in (1). Such a demonstration is critical for making a conclusion regarding the function of these genes.

(3) While Silpe et al. demonstrate the induction of latent phages *in vitro*, there are some conflicting observations in the paper regarding the effect of colibactin on phage-free strains. In figure 1, the authors show that there is no effect of colibactin on a phage-free isogenic strain; however, in Extended Data Figure 3, they do observe colibactin inhibiting prophage-free *S. aureus* strains by approximately two orders of magnitude. The authors should both provide an explanation for these differences with additional experimentation (for example, by addressing whether the increase in colibactin killing is dependent on phage island(s) in an isogenic background) and they should conduct experiments to understand the effect of colibactin on "phage-free *S. aureus*." For example, this strain-dependent difference in colibactin effect could reflect differences in the ability of *E. coli* and *S. aureus* to recover from DNA damage by the SOS response.

Minor concerns

(1) Lines 69-71 are confusing because they imply that there is no precedent for colibactin to affect bacteria; however, previous data described in the intro does imply that colibactin inhibits/kills bacteria. Please revise this sentence to make the motivation for the work clearer in the results.

(2) Are the ClbS homologs shown in Extended Data Figure 5 all of the blast hits? It is unclear from the main text and the figure legend if there were additional more divergent homologs of ClbS found. If there are more homologs, could the authors please include these in either a supplemental table or figure? This information would provide more evidence for the acquisition of resistance to colibactin.

Referee #2 (Remarks to the Author):

In this manuscript Silpe et al. identify colibactin as a prophage inducing natural product that relies on recA mediated cleavage of cI-like repressors for prophage activation. They show that the clbS immunity protein that protects against the action of colibactin or that exogenously added DNA is sufficient to block colibactin mediated prophage induction. Using bioinformatics they go on to show that some bacteria harbor clbS genes in absence of a full length pks gene cluster and that more broadly clbS orthologs are found in diverse gut bacteria outside of the usual suspects (i.e. Escherichia, Klebsiella, etc.). Overall, this is an elegant piece of bacterial genetics, which sheds light on the mechanism of colibactin in bacteria-bacteria interactions. Furthermore, there is the observation that clbS genes can be flanked by IS-like elements suggesting that clbS undergoes horizontal gene transfer, which if true will provide mechanistic insight in to how bacteria interact in polymicrobial communities and has important implications for microbiome research. This manuscript is well written, has an easy to follow narrative, and was a pleasure to read.

I have the following comments for the authors moving forward.

1. As much if not all of the pfu data shown are images, some being difficult to see clearly because of the figure contrast, I recommend that the authors also report their quantification of pfu/ml as bar graphs to accompany the images.
2. Many bacteria are polylysogenized. As it appears that only single lysogens were tested in this study, can the authors provide data showing that this mechanism can affect more than one genome encoded prophage.
3. It is stated that the effect of colibactin on prophage induction is taxonomically diverse. However, this is inferred indirectly from bioinformatics, or directly by only testing prophage induction by colibactin in *S. enterica* and *S. aureus*. I suggest that the authors dampen this statement, unless data can be provided to show this is much more broadly applied to diverse bacteria.
4. Were the clbS orthologs identified in the gut bacterial communities always associated with a pks gene cluster, or like in *E. coli* were there clbS orphans found as well?

Breck A. Duerkop

Referee #3 (Remarks to the Author):

This is a well-written manuscript by Silpe et al. details how bacterial product colibactin induces prophage in diverse bacteria, an effect that is prevented either by exogenous DNA addition or expression of resistance protein ClbS. The authors nicely demonstrate prophage-inducing effects using *E. coli*-lambda with co-culture systems including either colibactin-producing (pks+) *E. coli* or non-producers (pks-), and extend these findings to prophage induction in *Salmonella* and *Staphylococcus*. The authors also explore how resistance to colibactin mediated by ClbS may have

evolved from a self-protective mechanism to be spread and conserved in non-colibactin producing members of the microbiota to prevent prophage induction. They nicely demonstrate that ClbS/prophage homologs from Matakosakonia behave similarly in the E. coli system. This is an important finding that both improves understanding of the mechanisms of action of colibactin and proves a potential origin story for why this would have evolved, as well as offers some insights into complex community interactions. There are some lingering questions that remain, however, about how colibactin/ClbS are acting in culture as well as demonstrating these mechanisms of action a bit further beyond E. coli.

Major:

1. Is it possible to provide any information about the amount of colibactin present in supernatants/cultures? Is colibactin production/secretion altered by co-culture conditions and/or phage induction occurring in the culture?
2. Does co-culture that does not permit cell-cell contact (e.g. PMID 31936318 or some other system that permits constant exchange of secreted products but not direct interaction) mediate similar phage induction?
3. Where does ClbS act – is this also secreted or does it exclusively hydrolyze colibactin once colibactin is taken up by the cell? Or to say, can it mediate effects in trans? Do supernatants from ClbS+ bacteria have any effects towards limiting colibactin activity?
4. If ClbS is disrupted in colibactin-producing (pks+) E. coli that lack prophage, is there any phenotype/loss of competitive fitness?
5. Is it possible to disrupt the ClbS-like gene in native Metakosakonia or another of the identified non-E.coli microbes harboring homologs to look for +/- effects on phage induction in the native bacteria (phage could perhaps be quantified by qPCR or even just cfu assessed)? Or as an alternate approach, does addition of ClbS to Salmonella or Staph prevent induction of these native prophage by co-culture?

Minor:

1. Can Fig 1a gene cluster be further annotated/detailed to provide more information about the genes and their arrangement? Clbs, etc?
2. Do pks+ and MMC have additive effects on de-repression of phage replication (Fig 1e)?

Author Rebuttals to Initial Comments:

Editor comments:

I hope you are well. Your manuscript entitled "The gut bacterial natural product colibactin triggers induction of latent viruses in diverse bacteria" has now been seen by 3 referees, whose comments are attached below. While they find your work of potential interest, as do we, they have raised relevant concerns that in our view need to be addressed before we can consider publication in Nature.

The referees agree that the study is timely and intriguing, but in general, they felt that further work would be required to strengthen the insights relating to the generality of the findings, the mechanisms involved and their wider physiological relevance. In particular, and as mentioned by referee #1, we feel that expanding the study to demonstrate the action of colibactin in a setting resembling the complex environment of the gut, would strengthen the manuscript considerably. Should further experimental data allow you to address these criticisms, we would be happy to consider a revised manuscript (unless something similar has been accepted at Nature or appeared elsewhere in the meantime).

Thank you for handling our manuscript. A brief overview of the work we performed in the course of this revision is summarized below. Point-by-point responses to each of the reviewers' comments are noted as they appear in the review. Reviewers' comments are in black text and our responses are in red text.

As you will note in our responses, we have both increased the number of health/gut-relevant bacteria tested in our panel and expanded the conditions under which we test colibactin-dependent prophage induction. Specifically, we demonstrate that colibactin induces prophages residing in human-associated bacteria when they are grown in complex mouse-derived fecal communities. To demonstrate that prophage induction by colibactin can have important functional consequences, we also show that colibactin-mediated induction of a prophage in *C. rodentium* that carries a human disease-causing toxin (Shiga toxin, encoded by phage *stx*), results in upregulation of the toxin itself and an increase in its production. We believe these data provide compelling evidence that colibactin induces prophages in complex communities that resemble the gut, and that the effect of phage induction within these communities has potential health-relevant consequences (e.g. upregulation of phage-encoded human toxins).

In addition to evaluating prophage induction in complex communities, we obtained and tested one of the non-colibactin producing, *clbS*⁺ bacterial isolates from our initial panel to assess whether *clbS*-like proteins mediate resistance to colibactin-mediated prophage-induction in the organisms that encode them. As you will see in our revised submission, the bacterium we obtained (a human-derived isolate of *Escherichia albertii*) produces measurable phages in response to the known DNA-damaging agent mitomycin C but not after being co-cultured with colibactin-producing *E. coli*, indicating it is likely resistant to colibactin's effects. We perform additional experiments to implicate the ClbS-like protein from *E. albertii* in the observed protection.

Other revisions made to the manuscript in response to reviewer comments include: measuring colibactin production under our various co-culture conditions, determining additional details as to the site of ClbS action, elucidating the basis for species-specific sensitivity to colibactin, and providing additional details of our bioinformatic searches for ClbS homologs.

We thank the reviewers for their feedback, and we feel the changes we have made, described in detail below, have addressed their concerns. We also believe these changes have improved the manuscript and extended the impact of our work.

Referees' comments:

Referee #1 (Remarks to the Author)

Here, Silpe et al. implicate colibactin, an NRPS/PKS molecule produced by certain *E. coli*, in prophage activation within bacteria. This work potentially provides a mechanistic basis for the previously reported capacity of colibactin production to impact the gut microbiome bacterial community and act in an antibacterial manner. Such a role for colibactin is intriguing, in that it may provide an explanation for the production of a molecule previously found to act as a mutagenic agent of colonized humans and mice. The study is straightforward, the manuscript is well written, and the experiments are easily interpreted. My major concerns relate to the narrow scope of the work.

Major concerns

(1) Silpe et al. rely entirely on in vitro co-culture experiments in order to investigate prophage induction and antibacterial activity of colibactin on other bacteria. I find their data convincing that under such conditions, colibactin leads to prophage induction. This is not surprising given its known non-specific DNA crosslinking activity and the well characterized induction of prophage by such insults. However, there are many reasons why this activity might not be

observed (or relevant) in a more natural habitat, including the gut microbiome. For instance, colibactin activity is dependent upon highly electrophilic warhead structures that are readily quenched by extracellular molecules. Demonstrating prophage induction while measuring other potential concomitant consequences of colibactin on co-inhabiting bacteria in the context of an experiment involving a natural or model multispecies gut community is necessary in order to implicate the molecule in a meaningful way in bacterial interactions.

We thank the reviewer for the kind remarks on our work. We agree with the reviewer that demonstrating this effect in a complex community is an important objective.

First, with regards to the reactivity of colibactin – we agree with the reviewer's point that inactivation of colibactin in complex settings is likely to occur, however, we note that colibactin-specific genotoxic phenotypes have now been well established in a range of complex experimental setups, ranging from human organoids to mouse models. We therefore hypothesize that the same conditions that allow for colibactin-specific genotoxicity against mammalian cells in complex settings may also allow for colibactin activity toward bacteria in similar contexts.

As the connection between colibactin and phage induction is new, we agree with this reviewer that it would be valuable to understand whether the inducing effects we observe in two-way co-culture conditions also occur in communities of higher complexity. To address this issue, we have added several new experiments that demonstrate that colibactin activates prophages in a complex multispecies setting resembling the gut.

Specifically, included in the revised manuscript are data from four new sets of experiments in which we measure phage production and, in one case, its functional consequences in terms of toxin production, in mouse-derived fecal communities. These communities consist of complex mouse fecal samples to which we have added individual human-associated bacterial lysogens, including both strains from our initial submission and several additional human-/gut-relevant lysogens. The newly added bacteria include a human commensal isolate of *E. faecium* (harboring a previously uncharacterized temperate phage) and *C. rodentium* (harboring a Shiga toxin-encoding prophage, *stx_{2act}*). In each of the cases tested, we observed significant levels of phage production in these complex communities when lysogens were cultured with colibactin-producing as compared to non-producing *E. coli*. Using an ELISA assay to measure Shiga toxin under the same settings revealed that Stx2 toxin levels are significantly higher when in colibactin-producing *E. coli* are present in the community. These new findings, now appearing as Figures 3e-k, demonstrate that colibactin production leads to prophage-induction in mixed microbial communities that mimic the complexity of the intestinal setting. Furthermore, the finding that colibactin induces the *stx₂*-encoding prophage and leads to higher levels of the toxin itself in these communities demonstrates a direct, health-relevant consequence of colibactin-dependent phage induction.

(2) One interesting aspect of this study is the identification of *clbS* genes in *pks*-bacteria. Silpe et al. show that these genes, when over-expressed in *E. coli*, inhibit prophage induction. However, it remains unresolved whether these genes are able to perform this function when produced natively by the bacteria that harbor them and under conditions like those I mention in (1). Such a demonstration is critical for making a conclusion regarding the function of these genes.

We agree that it would be helpful to test whether expression of *ClbS* in native *clbS*-encoding *pks*-bacteria protects these organisms from prophage induction. While we were unable to obtain the isolate of *Metakosakonia* used as the basis for our heterologous expression experiments (Figure 3e in the initial submission), we were able to acquire a different human-

associated *clbS*-encoding organism from our panel, *Escherichia albertii* 07-3866. It is worth noting that *E. albertii* and *Metakosakonia* ClbS homologs, despite harboring different sequences, are encoded in similar relative genomic contexts (shown in Extended Data Figure 3e) downstream of predicted *recN* and *bamE* genes. This genomic arrangement (lacking the *clbS*) is also found in BW25113 *E. coli*.

We thus used *E. albertii* to assess whether its *clbS*-like gene provides protection from phage induction in this organism. Specifically, the revised manuscript now includes two pieces of new data consistent with the native *clbS* of *E. albertii* being functional (the relevant results are Figure 4e-i in the resubmitted version).

First, we experimentally determined that a prophage present in *E. albertii* is induced by mitomycin C (MMC) but not by co-culturing with *pks*⁺ *E. coli*, suggesting that the prophage in this isolate responds to DNA-damaging agents, in general, but is resistant to colibactin. This suggests the ClbS homolog is protective in this strain.

Second, while we were unable to genetically modify *E. albertii* (e.g. to cleanly delete *clbS*), we recombineered BW25113 *E. coli* (which, with the exception of the *clbS*, has ~90% nt identity in the adjacent up and downstream genes) to encode the *E. albertii* *clbS* sequence and its putative regulation on the *E. coli* chromosome. Although this is a heterologous expression experiment, the high degree of similarity in this genomic region between *E. albertii* and *E. coli* (specifically, surrounding two conserved genes, *recN* and *bamE*) should allow us to preserve the regulatory components that natively control *clbS* expression in *E. albertii* in the recombineered strain. When co-cultured with *pks*⁺ *E. coli*, we find that the recombineered *E. coli* strain harboring the *E. albertii* ClbS is significantly protected from the effects of colibactin as compared to the WT BW25113 *E. coli* (*clbS*⁻) strain.

Collectively, our new data on *E. albertii*, now appearing immediately after the existing data on *Metakosakonia* sp. (new Figure 4g-i), provide strong evidence that the *clbS* encoded by this non-colibactin producing bacterium is functional in limiting phage induction in the native organism.

We also note that we have performed an additional experiment to demonstrate the function of ClbS in organisms beyond *E. coli* in response to a separate comment raised by Referee 3 (comment 5). We include parts of that response below: We cloned and introduced a vector encoding the *pks*-associated *clbS* sequence from *E. coli* into *S. Typhimurium* and found that it provided similar resistance to prophage induction as we observed for *E. coli* (lower panels of the new Figure 4c). The results of this alternative gain-of-function experiment in *Salmonella* make a strong case that ClbS is capable of providing protection from prophage induction in *pks*⁻ bacteria.

(3) While Silpe et al. demonstrate the induction of latent phages in vitro, there are some conflicting observations in the paper regarding the effect of colibactin on phage-free strains. In figure 1, the authors show that there is no effect of colibactin on a phage-free isogenic strain; however, in Extended Data Figure 3, they do observe colibactin inhibiting prophage-free *S. aureus* strains by approximately two orders of magnitude. The authors should both provide an explanation for these differences with additional experimentation (for example, by addressing whether the increase in colibactin killing is dependent on phage island(s) in an isogenic background) and they should conduct experiments to understand the effect of colibactin on "phage-free *S. aureus*." For example, this strain-dependent difference in colibactin effect could reflect differences in the ability of *E. coli* and *S. aureus* to recover from DNA damage by the SOS response.

We agree with the reviewer than the discrepancy between colibactin-susceptibility of phage-

free

E. coli and *S. aureus* is interesting. Motivated by the possibility put forth by the reviewer – that the observed differences for colibactin may “reflect differences in the ability of *E. coli* and *S. aureus* to recover from DNA damage by the SOS response” – we exposed phage-free *S. aureus* and *E. coli* to varying doses of the known DNA damaging agent MMC. The new data, appearing as Extended Data Figure 5b, clearly show that phage-free *S. aureus* is significantly more susceptible to this standard DNA damaging agent than phage-free *E. coli*. Furthermore, a search of existing literature revealed that another class of DNA damaging agents, bis-indoles, have MICs ~1 order of magnitude lower in *S. aureus* than in *E. coli* (<https://dx.doi.org/10.1128%2FAAC.00309-16>). Overall, we believe that the differential effect we observe in phage-free *E. coli* as compared to phage-free *S. aureus* may simply be explained by a difference in sensitivity to DNA damaging agents between these organisms. In addition to including the new data, we have noted this point in the text (in the revised Discussion). We thank the reviewer for imploring us to test this possibility.

Minor concerns

(1) Lines 69-71 are confusing because they imply that there is no precedent for colibactin to affect bacteria; however, previous data described in the intro does imply that colibactin inhibits/kills bacteria. Please revise this sentence to make the motivation for the work clearer in the results.

Thank you for pointing this out. Our intention was to introduce the relevant literature on the subject and also to point out there has been no understanding of the basis for any antimicrobial activity noted for colibactin. For example, the idea that colibactin-producing *E. coli* inhibits other bacteria was posited (<https://dx.doi.org/10.1128%2FAAC.00130-16>), however, when tested, it had no effect on growth on all strains except *S. aureus*. As a result, and as the authors of that work noted, their data were not consistent with colibactin acting as a general antimicrobial agent. Thus, the mechanistic basis underlying colibactin's activity in an interbacterial setting has remained elusive. We have now modified the relevant lines noted by the referee to make the motivation of the present work clearer. Specifically, the new lines now read:

“Considering our limited understanding of the effects of colibactin in contexts outside of eukaryotic cells, we aimed to shed additional light on colibactin's activity and potential ecological roles in microbiomes by studying its effects on bacteria.”

(2) Are the ClbS homologs shown in Extended Data Figure 5 all of the blast hits? It is unclear from the main text and the figure legend if there were additional more divergent homologs of ClbS found. If there are more homologs, could the authors please include these in either a supplemental table or figure? This information would provide more evidence for the acquisition of resistance to colibactin.

The ClbS homologs previously shown in Extended Data Figure 5 (the relevant panels now appear as Extended Data Figure 3b and e in our revised manuscript) were not all of the BLAST hits. As requested, we have now included BLAST entries for all of the searches performed as a supplementary table (Supplementary Table 1 with tabs delineating the two different searches). Additional more divergent ClbS homologs were found in our expanded search. That particular search was performed using a BLASTp for the nearest 5,000 protein matches using the *E. coli* *pks*-associated ClbS as a query, and the search returned this maximum number of sequences (Extended Data Figure 3e and the second tab in Supplementary Table 1). The panel we selected for heterologous expression experiments encompasses proteins possessing 25-80% ID to the *E. coli* *pks*-associated ClbS. There were additional entries having <25% ID that will be interesting to test in the future; however, we chose for the purposes of the current work to focus on homologs with >25% amino acid ID.

Referee #2 (Remarks to the Author):

In this manuscript Silpe et al. identify colibactin as a prophage inducing natural product that relies on recA mediated cleavage of *ci*-like repressors for prophage activation. They show that the *clbS* immunity protein that protects against the action of colibactin or that exogenously added DNA is sufficient to block colibactin mediated prophage induction. Using bioinformatics they go on to show that some bacteria harbor *clbS* genes in absence of a full length *pks* gene cluster and that more broadly *clbS* orthologs are found in diverse gut bacteria outside of the usual suspects (i.e. *Escherichia*, *Klebsiella*, etc.). Overall, this is an elegant piece of bacterial genetics, which sheds light on the mechanism of colibactin in bacteria-bacteria interactions. Furthermore, there is the observation that *clbS* genes can be flanked by IS-like elements suggesting that *clbS* undergoes horizontal gene transfer, which if true will provide mechanistic insight in to how bacteria interact in polymicrobial communities and has important implications for microbiome research. This manuscript is well written, has an easy to follow narrative, and was a pleasure to read.

We are glad you enjoyed the work!

I have the following comments for the authors moving forward.

1. As much if not all of the pfu data shown are images, some being difficult to see clearly because of the figure contrast, I recommend that the authors also report their quantification of pfu/ml as bar graphs to accompany the images.

We agree that quantitation of PFU/mL is helpful. We have added an accompanying dataset with barplots and values for all PFUs generated in this work as a new supplementary table (Supplementary Table 5).

2. Many bacteria are polylysogenized. As it appears that only single lysogens were tested in this study, can the authors provide data showing that this mechanism can affect more than one genome encoded prophage.

Thank you for the suggestion. While we initially tested single lysogens for experimental simplicity, we agree that testing the effect of colibactin under more realistic scenarios (including polylysogenic bacteria) is important. To address this point (as well as comment 3 made by this reviewer below) we expanded our panel of bacteria tested for colibactin-mediated prophage induction. Namely, we obtained and tested a polylysogenic strain of *S. Typhimurium* (harboring BTP1 and Gifsy-1) in our co-culture setup. Using indicator strains specific to the two different phages present, our new data (Figure 3b in the revised manuscript) show that colibactin-producing *E. coli* induces both prophages from the polylysogen. We note that the magnitude of induction differs between the prophages, but this is not entirely unexpected as the two prophages also exhibit significant differences in basal (spontaneous) levels of induction presumably due to inherent genetic differences between them. Beyond testing the polylysogenic *S. Typhimurium* in our simple co-culture setup, we additionally tested this strain under anaerobic conditions in a mouse-derived complex fecal microbial community (new Figure 3e-g). We observe that colibactin-mediated prophage induction for *S. Typhimurium* also occurs in this complex setting.

3. It is stated that the effect of colibactin on prophage induction is taxonomically diverse. However, this is inferred indirectly from bioinformatics, or directly by only testing prophage induction by colibactin in *S. enterica* and *S. aureus*. I suggest that the authors dampen this statement, unless data can be provided to show this is much more broadly applied to diverse

bacteria.

Our data in the initial submission showed that multiple phage-bacteria systems respond to colibactin in co-culture, however, we agree with the referee's remark that we do not have sufficient evidence to assess the complete breadth of colibactin-mediated prophage induction. As the referee suggests, we have dampened every instance in which "taxonomically diverse" is used by deleting it or replacing it with terms like "different" or "multiple".

4. Were the *clbS* orthologs identified in the gut bacterial communities always associated with a *pks* gene cluster, or like in *E. coli* were there *clbS* orphans found as well?

The *clbS* orthologs identified in this study were found in bacterial isolate genomes not gut bacterial communities, and they were not always associated with a *pks* gene cluster. The distribution of the full length *pks* gene cluster, insofar as is currently known, is relatively limited and is thought to be produced by members of a handful of bacterial genera (including *Escherichia*, *Klebsiella*, *Frischella*, *Shigella*, and *Pseudovibrio*). While it is likely that other uncharacterized *pks*⁺ bacteria exist, the focus of our search led us specifically to *clbS* orthologs in *pks*⁻ bacteria.

Breck A. Duerkop

Referee #3 (Remarks to the Author):

This is a well-written manuscript by Silpe et al. details how bacterial product colibactin induces prophage in diverse bacteria, an effect that is prevented either by exogenous DNA addition or expression of resistance protein ClbS. The authors nicely demonstrate prophage-inducing effects using *E. coli*-lambda with co-culture systems including either colibactin-producing (*pks*⁺) *E. coli* or non-producers (*pks*⁻), and extend these findings to prophage induction in *Salmonella* and *Staphylococcus*. The authors also explore how resistance to colibactin mediated by ClbS may have evolved from a self-protective mechanism to be spread and conserved in non-colibactin producing members of the microbiota to prevent prophage induction. They nicely demonstrate that ClbS/prophage homologs from *Matakosakonia* behave similarly in the *E. coli* system. This is an important finding that both improves understanding of the mechanisms of action of colibactin and proves a potential origin story for why this would have evolved, as well as offers some insights into complex community interactions.

We thank the referee for the kind review!

There are some lingering questions that remain, however, about how colibactin/ClbS are acting in culture as well as demonstrating these mechanisms of action a bit further beyond *E. coli*.

Major:

1. Is it possible to provide any information about the amount of colibactin present in supernatants/cultures? Is colibactin production/secretion altered by co-culture conditions and/or phage induction occurring in the culture?

The theoretical maximum amount of colibactin generated in the cultures can be determined by quantifying the amount of an inactive co-product *N*-myristoyl-D-asparagine (known as the prodrug motif or scaffold) produced during maturation of the active genotoxin. To address this question, we used liquid chromatography-mass spectrometry to measure the amount of prodrug scaffold generated under 4 different conditions: *pks*⁻ in monoculture, *pks*⁺ in

monoculture, *pks*⁺ in co-culture with the phage-containing strain, and *pks*⁺ in co-culture with the phage-free strain. We found that the total amount of prodrug scaffold present was slightly lower in the co-culture experiments (20% less than that of the monoculture), and that there is no difference between the amount of colibactin produced in co-culture with a phage-containing or a phage-free strain, suggesting that phage induction does not alter colibactin production levels. These new data are now provided as Extended Data Figure 1d in the revised submission.

2. Does co-culture that does not permit cell-cell contact (e.g. PMID 31936318 or some other system that permits constant exchange of secreted products but not direct interaction) mediate similar phage induction?

We thank the reviewer for this suggestion. We tested this by limiting cell-cell contact between the *pks*^{+/-} *E. coli* and the lambda lysogen in a microtiter-based transwell assay (in which a filter with a 0.4 μm pore size divides each well). We did not observe phage induction during co-culture of *pks*⁺ *E. coli* and a lambda lysogen when the two strains were separated by the filter. We further confirmed that the transwell setup itself does not prohibit phage induction, in general, by showing that addition of MMC continues to induce phage production. We note that the lack of response to colibactin upon separating producing and target organisms is consistent with what others have previously observed in mammalian cell-focused studies, where separation of the colibactin-producing *E. coli* from target cells using a membrane abolishes activity (<https://dx.doi.org/10.1128%2FmBio.02393-17>). The new data now appear as Extended Data Figure 1c in the revised manuscript.

3. Where does ClbS act – is this also secreted or does it exclusively hydrolyze colibactin once colibactin is taken up by the cell? Or to say, can it mediate effects in trans? Do supernatants from ClbS⁺ bacteria have any effects towards limiting colibactin activity?

E. coli ClbS has previously been crystallized (<https://dx.doi.org/10.1021%2Fjacs.7b09971>). The current model for its activity based on this structural data is that ClbS destroys the cyclopropanerings of colibactin via a hydrolytic mechanism. As ClbS activity had not yet been examined in bacterial co-culture, we performed the experiment suggested by this reviewer, testing whether supernatants from ClbS⁺ bacteria limit colibactin activity. Our new data, which are presented in the resubmission as Extended Data Figure 3d, show that supernatants from ClbS⁺ cultures do not provide protection from colibactin in our setup. While there is still more to understand about the mode of action of ClbS, our data are consistent with the activity being confined to within the ClbS-expressing bacterial cell.

4. If ClbS is disrupted in colibactin-producing (*pks*⁺) *E. coli* that lack prophage, is there any phenotype/loss of competitive fitness?

It has previously been shown that deleting *clbS* from the *pks* gene cluster on a BAC results in a growth defect when the BAC is introduced into the common heterologous expression host *E. coli* DH10β, which is *recA*⁻ (<https://doi.org/10.1111/mmi.13272>). Since our experiments make use of WT *E. coli* (*recA*⁺), we measured the effect of deleting *clbS* on the fitness (growth) of the strains within our system. These data, presented as Extended Data Figure 3a, show that under the growth conditions in which the *recA*⁻ strain has a *pks*⁺/*clbS*⁻ induced growth defect, BW25113 (*pks*⁺/*clbS*⁻) (*recA*⁺) does not. This result indicates that deleting *clbS* in this *recA*⁺ strain does not result in a loss of fitness under the conditions tested. This matches our expectation, knowing that the presence of RecA in phage-free BW25113 *E. coli* allows for the recovery from sub-inhibitory DNA damage via the SOS response, whereas *recA*⁻ strains are known to be hypersensitive to DNA damage (<https://doi.org/10.1080/009841099157683>).

5. Is it possible to disrupt the ClbS-like gene in native *Metakosakonia* or another of the identified non-*E. coli* microbes harboring homologs to look for +/- effects on phage induction in the native bacteria (phage could perhaps be quantified by qPCR or even just cfu assessed)? Or as an alternate approach, does addition of ClbS to *Salmonella* or *Staph* prevent induction of these native prophage by co-culture?

Thank you for these ideas. The proposal to test the function of ClbS in a native *clbS*⁺/*pks*⁻ organism was also raised by Reviewer 1 (point 2). Parts of this response are duplicated from that one: While we were unable to obtain the isolate of *Metakosakonia* used as the basis for our heterologous expression experiments (Figure 3e in the initial submission), we were able to acquire a different human-associated *clbS*-encoding organism from our panel, *Escherichia albertii* 07-3866. It is worth noting that *E. albertii* and *Metakosakonia* ClbS homologs, despite harboring different sequences, are encoded in similar relative genomic contexts (shown in Extended Data Figure 3e) downstream of predicted *recN* and *bamE* genes. This genomic arrangement (lacking the *clbS*) is also found in BW25113 *E. coli*.

We thus used *E. albertii* to assess whether its *clbS*-like gene provides protection from phage induction in this organism. Specifically, the revised manuscript now includes two pieces of new data consistent with the native *clbS* of *E. albertii* being functional (the relevant results are Figure 4e-i in the resubmitted version).

First, we experimentally determined that a prophage present in *E. albertii* is induced by mitomycin C (MMC) but not by co-culturing with *pks*⁺ *E. coli*, suggesting that the prophage in this isolate responds to DNA-damaging agents, in general, but is resistant to colibactin. This suggests the ClbS homolog is protective in this strain.

Second, while we were unable to genetically modify *E. albertii* (e.g. to cleanly delete *clbS*), we recombineered BW25113 *E. coli* (which, with the exception of the *clbS*, has ~90% nt identity in the adjacent up and downstream genes) to encode the *E. albertii* *clbS* sequence and its putative regulation on the *E. coli* chromosome. Although this is a heterologous expression experiment, the high degree of similarity in this genomic region between *E. albertii* and *E. coli* (specifically, surrounding two conserved genes, *recN* and *bamE*) should allow us to preserve the regulatory components that natively control *clbS* expression in *E. albertii* in the recombineered strain. When co-cultured with *pks*⁺ *E. coli*, we find that the recombineered *E. coli* strain harboring the *E. albertii* ClbS is significantly protected from the effects of colibactin as compared to the WT BW25113 *E. coli* (*clbS*⁻) strain.

Specific to this comment (not mentioned by Reviewer 1), we wish to note that we also performed the “alternative approach” experiment proposed by this Reviewer. In particular, we cloned and introduced a vector encoding the *pks*-associated *clbS* sequence from *E. coli* into *S. Typhimurium* and found that it provided similar resistance to prophage induction as we observed for *E. coli* (lower panels of the new Figure 4c). We hope that our efforts to test the function of ClbS in native organisms (the above-mentioned *Metakosakonia* sp. and *E. albertii* experiments) paired with the results of this alternative gain-of-function experiment in *Salmonella* make for a strong case that ClbS is likely expressed and capable of providing protection from prophage induction in *pks*⁻ bacteria.

Minor:

1. Can Fig 1a gene cluster be further annotated/detailed to provide more information about the genes and their arrangement? Clbs, etc?

Per the referee's suggestion, we have included a more detailed version of the *pks* gene cluster in the revised manuscript as Figure 1a.

2. Do *pks*⁺ and MMC have additive effects on de-repression of phage replication (Fig 1e)?

The known reactivity of MMC and colibactin with DNA, combined with the results of our current work, suggest that colibactin activates the bacterial SOS response in a similar RecA-dependent manner as other DNA-damaging agents, such as MMC. As a result, we suspect that under conditions in which MMC and colibactin are present in individually sub-saturating amounts, the effect on phage de-repression should be additive. We tested this experimentally using a SOS-inducible reporter, and under our standard setup, indeed observed this additivity when MMC was added (see Appendix Data 1, below). It is worth noting that controlling the amount of colibactin produced in these experiments is technically challenging. Unlike MMC, which is available as a pure compound and can be added at a single time point, colibactin must be continuously produced from the co-culture. Moreover, different factors such as initial culture density, mixing ratios, time of MMC introduction, may influence the dynamics of de-repression of phage replication in ways that are difficult to predict. Because of these challenges, we have not systematically investigated this preliminary observation, and we are therefore wary to definitively conclude this additive effect is operative. As such, we have not incorporated this point into the revised text.

Appendix Data 1: Relative light units (RLU) produced from a SOS-inducible bioluminescent reporter co-cultured with *pks*⁻ *E. coli*, *pks*⁻ *E. coli* and MMC, or *pks*⁺ *E. coli* and MMC. MMC was added at a final concentration of 100 ng mL⁻¹. Data represented as mean ± SD with n = 3 biological replicates.

Reviewer Reports on the First Revision:

Referee #1 (Remarks to the Author):

The authors have substantially strengthened the manuscript with new findings that explore the significance of colibactin-linked prophage induction in vivo. In particular, new data demonstrating activation of prophage within strains added to mouse-derived fecal communities bolster the central claim of the study. These experiments do not ultimately provide insights into the impact of colibactin prophage induction activity within a bacterial community, but they do provide compelling evidence that the phenomenon can occur. Other additions to the manuscript include the demonstration that a *clbS* gene derived from a *pks*⁻ bacterium can inhibit prophage induction in a near-native expression scenario and the characterization of *S. aureus*'s general sensitivity to DNA damaging agents. These experiments largely address my concerns.

Referee #2 (Remarks to the Author):

The authors have done a wonderful job addressing my previous comments. I have no further comments.

Referee #3 (Remarks to the Author):

The authors did an admirable job of addressing my prior questions/concerns, and succeeded in further extending their findings in a compelling fashion.

Author Rebuttals to First Revision:**Editor comment:**

I hope you are well. Your manuscript entitled "The gut bacterial natural product colibactin triggers induction of latent viruses in diverse bacteria" has now been seen again by our referees, and in the light of their advice I am delighted to say that we can in principle offer to publish it. First, however, we would like you to revise your paper to address the points made by the referees, and to make some editorial changes to your paper so that it is as brief as possible and complies with our Guide to Authors (<https://www.nature.com/nature/for-authors>). No peer reviewed data should be removed altogether when making these changes.

Thank you very much for your time and care in handling our manuscript. Our updated submission package contains all files of the manuscript according to *Nature's* final submission format. Below are our responses to the referees (red). Importantly, in the process of preparing the revised submission, no new data were added, and no prior data were excluded or modified.

Referees' comments:**Referee #1 (Remarks to the Author):**

The authors have substantially strengthened the manuscript with new findings that explore the significance of colibactin-linked prophage induction in vivo. In particular, new data demonstrating activation of prophage within strains added to mouse-derived fecal communities bolster the central claim of the study. These experiments do not ultimately provide insights into the impact of colibactin prophage induction activity within a bacterial community, but they do provide compelling evidence that the phenomenon can occur. Other additions to the manuscript include the demonstration that a *clbS* gene derived from a *pks*-bacterium can inhibit prophage induction in a near-native expression scenario and the characterization of *S. aureus*'s general sensitivity to DNA damaging agents. These experiments largely address my concerns.

Thank you for your time and attention on this submission. We are happy that you enjoyed the latest additions, and we wholeheartedly agree that they improved the piece.

Referee #2 (Remarks to the Author):

The authors have done a wonderful job addressing my previous comments. I have no further comments.

Thank you for the kind words. We are happy that we were able to address your concerns and greatly appreciated your fair and thorough evaluation of the work during the process.

Referee #3 (Remarks to the Author):

The authors did an admirable job of addressing my prior questions/concerns, and succeeded in further extending their findings in a compelling fashion.

Thank you for your thoughtful feedback throughout this process. We are glad to be able to answer your questions. The additional experiments we performed in response to your comments ended up expanding our own understanding of the topic, something we know future readers will appreciate.